



# 1 Measurement report: Atmospheric new particle formation in a peri-

# 2 urban site in Lille, Northern France

Suzanne Crumeyrolle[1], Jenni SS Kontkanen[2,3], Clémence Rose[4], Alejandra Velasquez Garcia[1,5], Eric
Bourrianne[1], Maxime Catalfamo[1], Véronique Riffault[5], Emmanuel Tison[5], Joel Ferreira de Brito[5], Nicolas
Visez[6], Nicolas Ferlay[1], Frédérique Auriol[1], Isabelle Chiapello[1]
[1] Univ. Lille, CNRS, UMR 8518 Laboratoire d'Optique Atmosphérique (LOA), 59000 Lille, France
[2] CSC - IT Center for Science, Espoo, Finland
3 Institute for Atmospheric and Earth system Research, University of Helsinki, Helsinki, Finland
[4] Laboratoire de Météorologie Physique, LaMP-UMR 6016, CNRS, Université Clermont Auvergne, 63178, Aubière, France
[5] IMT Nord Europe, Institut Mines-Télécom, Univ. Lille, Centre for Energy and Environment, F-59000 Lille, France
[6]Univ. Lille, CNRS, UMR 8516 - LASIRE - LAboratoire de Spectroscopie pour les Interactions, la Réactivité et
l'Environnement, F-59000 Lille, France.
*Correspondence to*: Suzanne Crumeyrolle (suzanne.crumeyrolle@univ-lille.fr)
**Abstract.**
Formation of Ultrafine particles (UFPs) in the urban atmosphere is expected to be less favored than in the
rural atmosphere due to the high existing particle surface area acting as a sink for newly-formed particles
Despite the large condensation sink (CS) values, previous comparative studies between rural and urban
site reported higher frequency of new particle formation (NPF) events over urban sites in comparison to
background sites as well as higher particle formation and growth rates attributed to the higher
concentration of condensable species. The present study aims to better understand the environmental
factors favoring, or disfavoring, atmospheric NPF over Lille, a large city North of France and to analyze
their impact on particle number concentration using a long-term dataset (4 years : 1st July 2017 to 31st
December 2020).





The results highlight a strong seasonal variation of the NPF occurrences with a maximum observed during
spring (27 events) and summer (53 events). It was found that high temperature (T > 295K), low RH (RH<
45%) and high solar radiation are ideal to observe NPF events over Lille. Relatively high values of
condensation sink (CS ~$2.10^{-2}$ $s^{-1}$) are reported during event days suggesting that high CS does not inhibit
the occurrence of NPFt over our site. Moreover, the particle Growth Rate ($GR_{15.7-30nm}$) was positively
correlated with the temperature most probably linked to the higher emissions of precursors. Finally, the
nucleation strength factor (NSF) was calculated to highlight the impact of those NPF events on particle
number concentrations. $NSF_{15.7-100}$ reaches a maximum of 4 in summer, indicating an enormous
contribution of NPF events to particle number concentration at this time of the year.
**1    Introduction**
New Particle Formation (NPF) leads to the formation of a large number of sub-20nm particles that will
contribute to the levels of fine particles observed in ambient air. The latter can have adverse effect on
human health as they can penetrate deeply into the pulmonary system (Clifford et al., 2018; Ohlwein et
al., 2019). The freshly-formed particles then grow to larger sizes, from a few nm in particle diameter up
to sizes (Dp > 100 nm) at which they may act as cloud condensation nuclei (CCN, (Pierce and Adams,
2009; Ren et al., 2021; Rose et al., 2017; Spracklen et al., 2006). NPF events have been observed around
the world (Kerminen et al., 2018; Kontkanen et al., 2017; Kulmala et al., 2004) in various environments
from the boundary layer (BL) at urban locations (Ganguly and Jayaraman, 2006; Roig Rodelas et al.,
2019; Tuch et al., 2006; Wehner and Wiedensohler, 2003) as well as remote polar background areas
(Dall'Osto et al., 2018) but also within the free troposphere (Rose et al., 2015b, 2015a). NPF events are
typically associated to a photochemical origin, thus occurring mostly during daytime (Kulmala et al.,
2014), with some scarce events being observed during nighttime (Roig Rodelas et al., 2019; Salimi et al.,

47    2017).


NPF occurrence depends on various factors including precursor emission strength, number concentration
of pre-existing aerosol population, meteorological parameters (in particular solar radiation, temperature



and relative humidity) and oxidation capacity of the atmosphere (Kerminen et al., 2018). Differences were
found in both the seasonality and intensity of NPF events according to the site type (urban, traffic, regional
background, rural, polar, high altitude (Dall'Osto et al., 2018; Sellegri et al., 2019). This variability seems
to be related to the environmental conditions that are specific to each location, which makes it hard to
draw general conclusions on the conditions that trigger NPF events (Berland et al., 2017; Bousiotis et al.,
2021). However, Nieminen et al. (2018) highlighted a common seasonal occurrence of NPF during spring
and summer using datasets from 36 continental sites worldwide.
The formation and growth of initial clusters to detectable sizes ($Dp > 1\text{-}3$ nm) compete with their
simultaneous removal from the ultra-fine particle mode by coagulation with pre-existing particles
(Kerminen et al., 2001; Kulmala, 2003). Because of this, the number concentration of particles smaller
than 20 nm has been observed to be anti-correlated with the aerosol volume and mass concentration over
rural area in Northern Italy (Rodríguez et al., 2005). Indeed, the total aerosol volume is rather small during
nucleation events (Kerminen et al., 2018; Rodríguez et al., 2008). While the negative effect of increased
pre-existing particle surface area (often described with the condensation sink, CS) on the occurrence of
these events is widely accepted (Kalkavouras et al., 2017), yet cases are found when NPF events occur
on days with higher CS compared to average conditions (Größ et al., 2018; Kulmala et al., 2017).
A recent study (Bousiotis et al., 2021) using large datasets (16 sites) over Europe (6 countries) highlighted
that solar radiation intensity, temperature, and atmospheric pressure had a positive relationship with the
occurrence of NPF events at the majority of sites (exceptions were found for the southern sites), either
promoting particle formation or growth rate. Indeed, solar radiation is considered one of the most
important factors in the occurrence of NPF events, as it contributes to the production of NPF precursors.
Higher temperatures are considered favorable for the growth of newly formed particles as they can be
linked to increased concentrations of organics vapor (Wang et al., 2013) that support particle but also
reduce the stability of the initial molecular clusters (Deng et al., 2020; Kurtén et al., 2007).
The wind speed, on the other hand, has presented variable effects on the occurrence of NPF events results,
appearing to depend on the site location rather than their type (Bousiotis et al., 2021). Additionally, the
origin of the incoming air masses plays a very important role, since air masses of different origins have
different meteorological, physical and chemical characteristics. Therefore, the probability of NPF event





occurrence at a given location and time depends not only on local emissions, but also on long range
transport (Sogacheva et al., 2007, 2005; Tunved et al., 2006) and on synoptic meteorological conditions
at the European scale (Berland et al., 2017).
Formation of new particles in the urban atmosphere is expected to be less favored than in the rural
atmosphere due to the high existing surface area of particles acting as a sink for freshly-formed particles.
Despite the large CS values, previous comparative studies between rural and urban site reported higher
frequency of NPF events (Peng et al., 2017) over urban sites in comparison to background sites as well
as higher particle formation and growth rates (Nieminen et al., 2018; Salma et al., 2016; Wang et al.,
2017) attributed to the higher concentration of condensable species. This study presents the first
observations of new particle formations over Lille, a large city in the north of France. Based on a multi-
annual dataset (2017-2020), the frequency and intensity of the events are analyzed aiming to better
constrain the favorable and unfavorable conditions.

**2    Materials and methods**
The ATOLL (ATmospheric Observations in LiLLE) station is located in Villeneuve d'Ascq, Northern
France (50.6114 N, 3.1406 E), and only 4 km away from the city center of Lille, which is the core of the
metropolis (Métropole Européenne de Lille, more than 1.1 million inhabitants) to which Villeneuve
d'Ascq belongs. Observations such as low Single Scattering Albedo (SSA) values (0.75 on average within
the $PM_1$ fraction, Velasquez-Garcia et al., under review) and large particle number concentrations (6140
$cm^{-3}$ on average) suggest that aerosol measurements performed at ATOLL aerosol conditions are
comparable to GAW sites classified as urban (Laj et al., 2020). ATOLL is also part of ACTRIS (Aerosols,
Clouds, and Trace gases Research InfraStructure Network, http://www.actris.net) program,
complementing the high-quality long-term atmospheric data in Northern France. This station is under the
influence of many anthropogenic sources, e.g. road traffic, residential sector, agriculture and industries
(Chen et al., 2022), as well as maritime emissions, and more episodically under the influence of events of
aged volcanic plumes and desert dust (Bovchaliuk et al., 2016; Mortier et al., 2013).

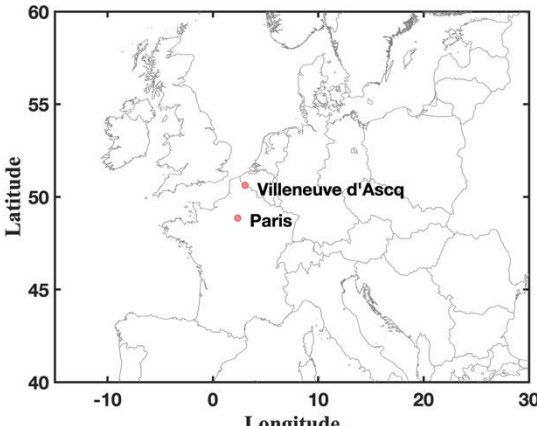
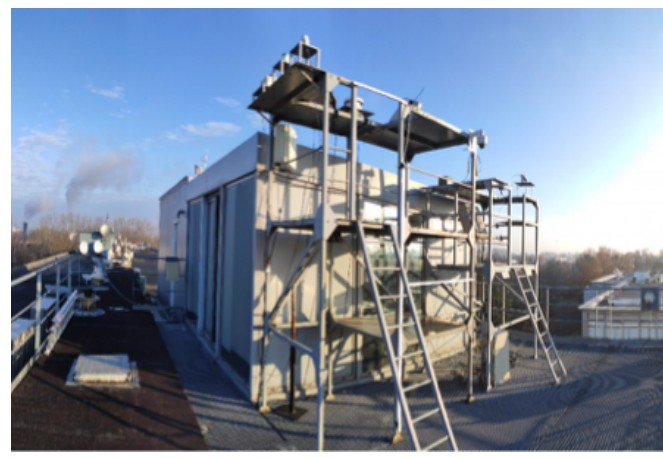

Figure 1 : ATOLL location in Villeneuve d'Ascq (Northern France) and picture of the station on the rooftop of the University of Lille P5 building (ⓒ LOA).


A large set of *in-situ* and remote sensing instruments are implemented in ATOLL to characterize physical,
chemical, optical and radiative properties of particles and clouds. *In-situ* instruments have independent
sampling stainless-steel lines located at least 1 meter above the roof top and equipped either with $PM_1$
cyclone or $PM_{10}$ inlet. The measurements used for that study were performed between 1st July 2017 and
31st December 2020. The instruments use in this study focused on aerosol properties including number
size distributions, chemical composition, and optical properties, and details are described below.
The Scanning Mobility Particle Sizer (SMPS) measures particle number size distribution between 15.7-
800 nm. The SMPS system consisted of a TSI model 3775 condensation particle counter, a TSI 3082 –
type differential mobility analyzer (DMA) as described by (Villani et al., 2007) and a Nickel aerosol
neutralizer (Ni-63 95MBq). The sheath flow rate was controlled with a critical orifice in a closed loop
arrangement (Jokinen and Mäkelä, 1997). Typically, the scan time was chosen to be 300 seconds. To take
into account the multiple charge effect and the losses through diffusion, particle concentrations were
corrected using the equation given by the manufacturer specifications (AIM 10.2.0.11).



Accordingly, aerosol number size distribution data from the SMPS measurements were used to classify
individual days as NPF event, undefined or non-event days. The classification followed the procedure
presented in (Dal Maso et al., 2005) following the decision criteria based on the presence of fine particles
(Dp < 25 nm) and their consequent growth to Aitken mode. Briefly, event days are identified when sub-
25nm particle formation and growth are observed. On non-event days nucleation mode is absent. Finally,
undefined days are the days when sub-25nm particles are observed but do not grow subsequently or last
less than an hour.
SMPS dry (using a Nafion) particle number size distributions were also used for CS and growth rate (GR)
calculations. The CS estimates the loss rate of the condensable vapors (Kulmala et al., 2001) which were
assumed to have molecular properties similar to sulfuric acid for CS calculation (Dal Maso et al., 2005).
A high CS indicates the presence of large surface area of aerosol particles onto which NPF precursors can
condensate and particles can coagulate as well. The particle GR was calculated based on the maximum-
concentration method described in (Kulmala et al., 2012). First, the NPF starting time was identified when
the newly formed mode was observable in the first bins of the SMPS (15.7 nm) and the time of peak
concentrations for particles with diameter of 30 nm ($N_{30}$) during NPF were manually identified. Particle
$GR_{15.7 - 30}$ was then calculated by linear regression of particle size vs. time span from the NPF start until
time when $N_{30}$ reaches a maximum.
Absorption coefficients ($\sigma_{abs}$) were continuously measured with a seven-wavelength aethalometer (AE33,
Magee Scientific Inc., Cuesta-Mosquera et al., 2020). According to ACTRIS current guidelines
(https://actris-ecac.eu/particle-light-absorption.html), $\sigma_{abs}$ coefficients at each wavelength have been
recalculated by 1) multiplying equivalent Black Carbon (eBC) by the mass-specific absorption coefficient
(MAC) and then 2) dividing by the suitable harmonization factor to account for the filter multiple
scattering effect, i.e. 2.21 (M8020 filter tape) in 2017 and 1.76 (M8060 filter tape) afterwards. The
aethalometer samples at 5 L.min$^{-1}$ downstream a PM$_1$ cyclone (BGI SCC1.197, Mesa Labs). The spectral
dependency of $\sigma_{abs}$ was used to determine the contributions of traffic (fossil fuel - FF) and Wood Burning
(WB) to eBC via a source apportionment model (Sandradewi et al., 2008).
Meteorological data including temperature, water vapour mixing ratio, and solar radiation were also
measured every minute at the sampling site using a weather station (DAVIS Inc weather station, Vantage



Pro 2) and a set of Kipp and Zonen pyranometer (CM22), pyrheliometer (CH1) and pyrgeometer (CGR4).
A skyimager (Cloudcam, CMS) was also used to estimate the sky cloudiness (Shukla et al., 2016).
Three-day air mass backtrajectories of air masses arriving at ATOLL at half the boundary layer height
between July 1, 2017 and December 31, 2020 were computed every hour using the Hybrid Single-Particle
Lagrangian Integrated Trajectory (HYSPLIT version 5.1.0) transport and dispersion model from the
NOAA Air Resources Laboratory (ARL) (Rolph et al., 2017; Stein et al., 2015) and meteorological input
from the Global Data Assimilation System (GDAS) at 1×1° resolution, resulting in 30719
backtrajectories.

## 3    Results

### 3.1    NPF event frequency and Growth rate

The seasonal distribution of NPF events at the ATOLL site is displayed in Figure 2. SMPS missing data
(in Figure 2) are about 40% from January to April due to the yearly calibrations at the manufacturer
premises and few laboratory campaigns (Oct 2018 – Jun 2019). Over the 4 years of measurements (2017-
2020), 96 (11%) days were classified as NPF events, 355 (40%) as undefined days and 432 (49%) as non-
event days. One can also note that most of the NPF events identified at the ATOLL site were observed
during spring (March-April-May, 27 events corresponding to 15% of the days when observations were
available during this season) and summer (June-July-August, 53 events corresponding to 19%) with a
maximum observed in June consistent with a previous study over central Europe (Dall´Osto et al., 2018).
During winter, the number of events is extremely limited (only one event observed in February). In the
following sections, only observations from spring and summer seasons will be discussed due to the low
representativeness of NPF events in fall (n=15) and winter (n=1). Moreover, the undefined event days are
seen all year round (frequency around or larger than 20%) with a clear peak in August (frequency at
62.5%) consistent with observations over boreal forest where undefined days were also observed to be
most frequent in early fall (Mazon et al., 2009).
Using long-term measurements from 36 sites (polar, rural, high altitude, remote, urban), Nieminen et al.,
(2018) reported an annual NPF frequency below 15% for half of the sites (18 sites from all types) and



occasionally over 30% for 10 sites. Moreover, they highlighted a seasonal variation of NPF occurrence
with larger (lower) frequency, about 30 % (10%), during spring (winter). Frequency analysis of NPF
occurring only over urban or anthropogenically influenced sites show large site-to-site differences for all
seasons. Indeed, NPF occurrence frequencies are varying from 20% (Helsinki in Finland, Sao Paulo in
Brazil) to 80% (Beijing in China, Marikana in South Africa) during spring and from 7% (Helsinki) to
78% (Marikana) during winter. Yearly average of NPF occurrence frequencies are between 11%
(Helsinki) and more than 60% (Beijing and Marikana).

The ATOLL event frequency (seasonal variation and values) is similar to observations performed in Paris
(Dos Santos et al., 2015) while the frequency of undefined and non-event days are quite different. Indeed,
in Paris the non-event frequency is larger than 60% except in July and August whereas over ATOLL the
non-event frequency shows a clear seasonal pattern with lower frequency (<40%) from April to August.
Moreover, undefined event frequency in Paris shows a minimum (<5%) in May and June and remains
quite steady during the rest of the year (around 20%). One can note that the frequency of undefined events
(also considered as failed events) is much higher over ATOLL all year long with an average of 40% while
it remains below 40% over the boreal forest. The frequency of undefined events observed at ATOLL is
clearly larger than the frequencies observed over more polluted site (Paris) and a pristine site (Boreal
forest). This might show that ATOLL is under the influence of air masses or particle and precursor sinks
that favor the burst of UFP.




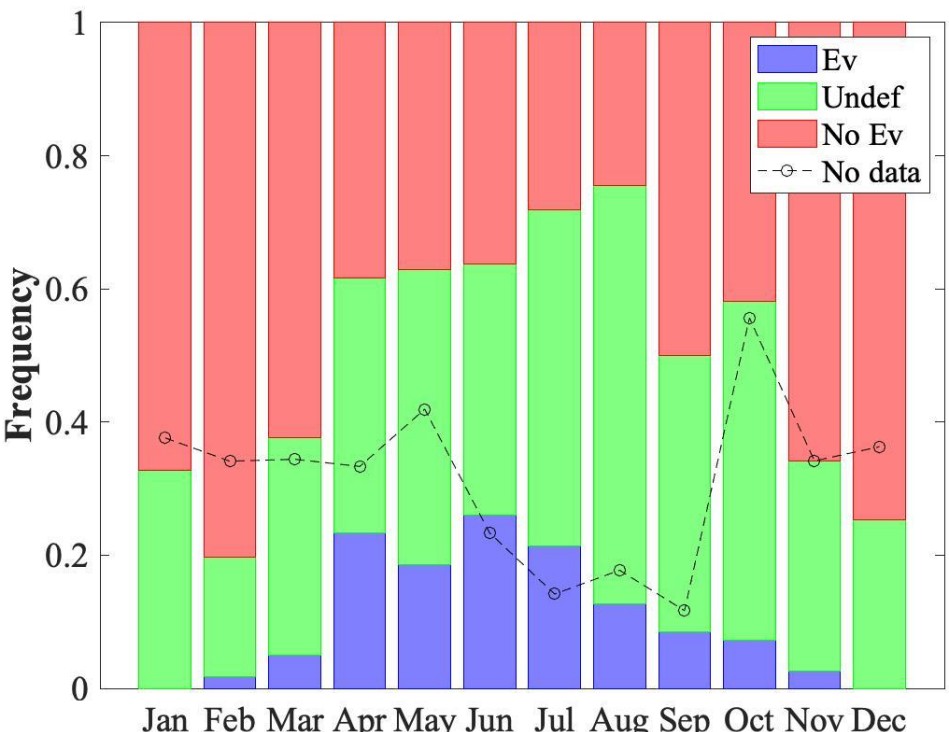

**Figure 2 : Seasonal distribution of event days (blue), undefined days (green), and non-event (red) days at the ATOLL station, Lille, France, during 2017–2020. Days with missing data are excluded from the total number of days per month and the frequency of missing data are indicated with the black circles.**


**3.2    Aerosol dry size distribution**
Median daily contour plots of the particle number size distributions (PNSD) obtained from the SMPS are
shown in Figure 3 separately for NPF event, undefined and non-event days observed during the warm
period (only spring and summer). Atmospheric NPF and subsequent particle growth are seen in Figure 3a
as an emergence of new aerosol particles with small diameter followed by the growth of these particles
into larger sizes. If this phenomenon is taking place regionally (few tens of km in radius), a so called
'banana plot' is observed in particle number size distributions as a function of time at a fixed location.





The time evolution of the "median NPF day" (Figure 3a) displays a similar growth pattern for newly
formed particles than for individual NPF event days (See supplementary materials). Indeed, one can
clearly see a UFP mode appearing from 10:00 to 15:00 (UTC) and growing during the rest of the day.
The NPF starting time and the growth rate will be discussed in the following section. By 23:00 UTC, the
newly formed particles reach an average diameter of 50 nm, similar to the geometric diameter of the mode
of the pre-existing particles observed during the morning (00:00 – 08:00). The "median undefined day"
(Figure 3b) highlights a burst of UFP from 10:00 to 15:00 (UTC) that is not growing and does not last
during the whole afternoon.   The behavior of the median is again similar to the individual undefined
events observed during this period. The "median non-event day" (Figure 3c) shows no sign of particle
growth, as expected.

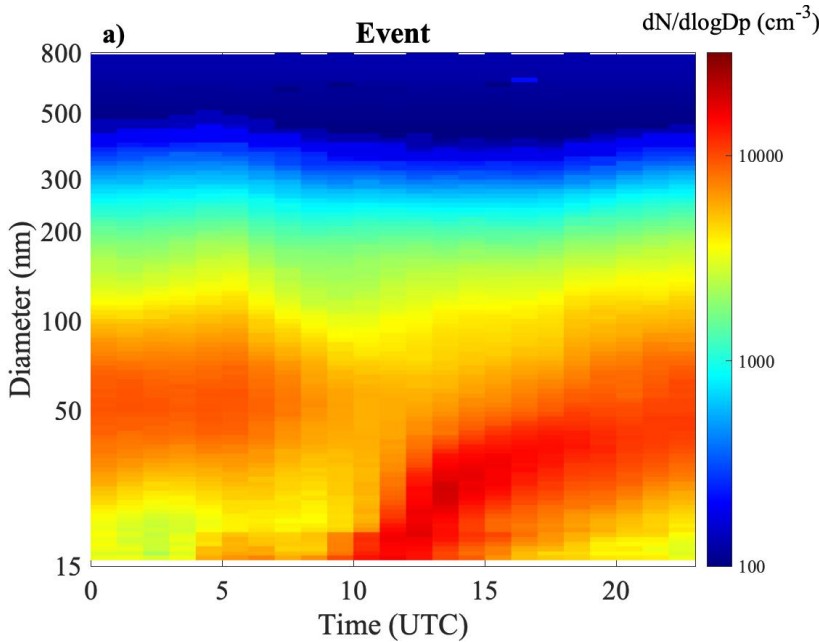

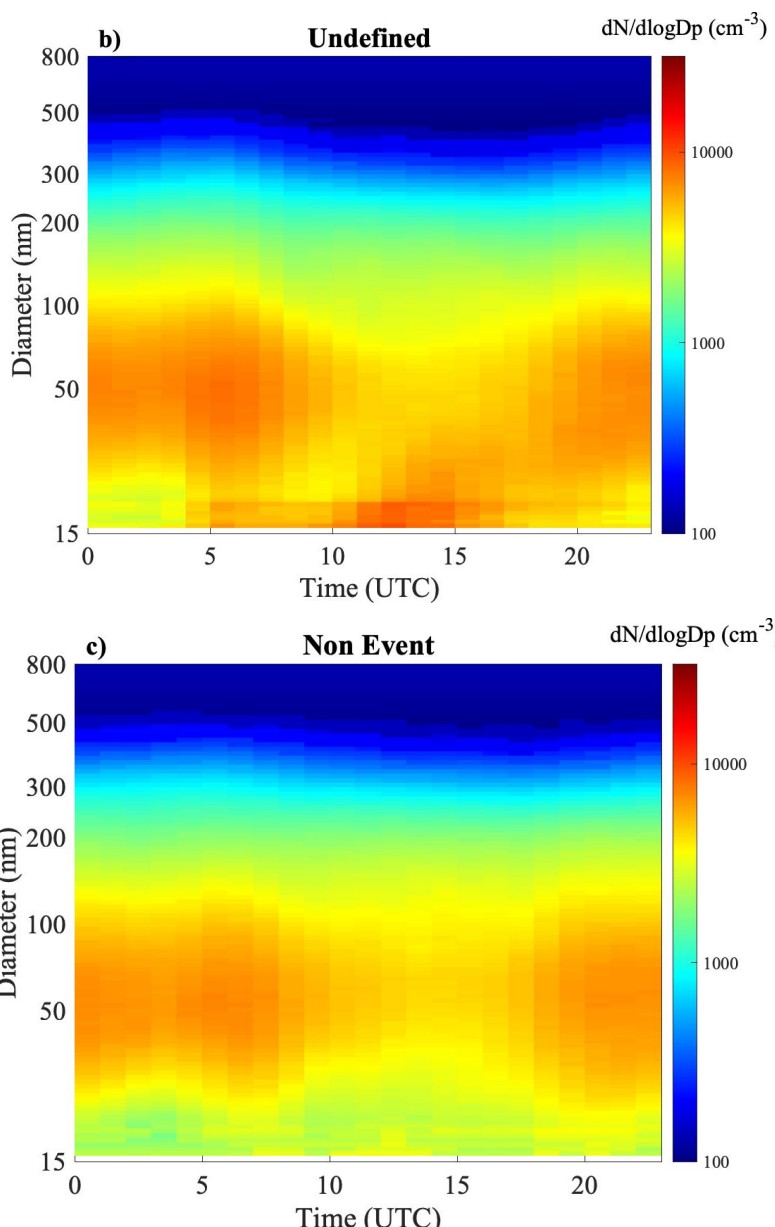

Figure 3 : Median particle number size distribution (15.7 nm<Dp<800 nm) observed during NPF event (a), undefined (b) and non-event (c) days in spring and summer from 2017-2020.



**3.3    Starting time and Growth rate**
Figure 4 shows the monthly variation of the starting time and growth rate of each event observed at
ATOLL. Most NPF events observed in ATOLL were observed to start between 09:00 and 14:00 local
time (74%), with fewer events in the early morning (07:30 - 09:00, 6%) and late afternoon (15:00 and
19:30, 20%). NPF starting time as well as GR strongly depend on the month during which the event is
observed. Indeed, the NPF starting time becomes earlier during the colder period and reaches a minimum
in June (around 08:20). No NPF event were observed after 16:00 in summer. During spring and fall, the
average NPF starting time varies between 10:00 and 19:00. Nocturnal events are rarely observed, with
only one occurrence in August. The start time monthly variability is linked to sunrise and sunset times.
In the following section, a link between the total solar irradiation and NPF occurrence will be examined.
The event ending time was determined as the time when the growth of the freshly formed particles was
over, i.e. when the diameter reached the diameter of the pre-existing particles. The duration of nucleation
events, at ATOLL, was then estimated and varies from an hour up to 28 hours. On average, NPF duration
is shorter from May to August (around 8 hours) and increases up to around 13 hours on average during
colder months (March and November). This seasonal behavior could be due to the presence of availability
of condensable vapors, air mass origin, and environmental conditions favorable to NPF events (see section
33    3.2).

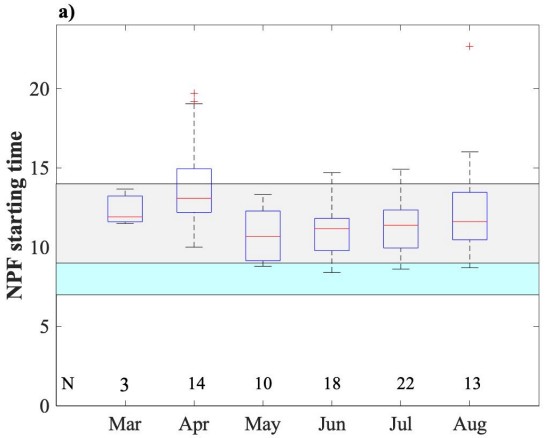

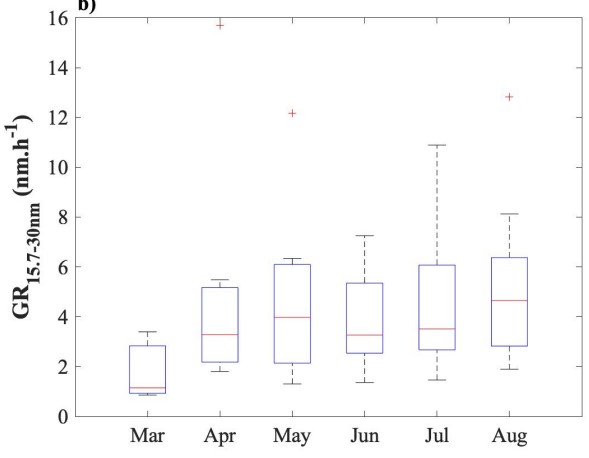




**Figure 4 : Monthly variation of new particle formation starting time (a) and their Growth Rate (GR$_{15.7-30nm}$) at the ATOLL station during 2017–2020. The grey area represents the period, from 09:00-14:00, when most of the NPF events occur. The blue area corresponds to the period before the NPF onset (07:00- 09:00). N represents the number of events observed per month.**

The growth rate values (GR$_{15.7-30nm}$) observed at ATOLL lie within 0.8 to 15.7 nm.h$^{-1}$ and show a strong monthly variation with the lowest values observed in spring and fall and the largest ones observed during summer (Figure 4). GR$_{15.7-30nm}$ values were in addition plotted as a function of temperature for all years and seasons in Figure 5, which highlights that below 20°C, GR$_{15.7-30nm}$ values are lower than 6 nm.h$^{-1}$, while, under warmer conditions (T >20 °C), GR$_{15.7-30nm}$ reach values up to 16 nm.h$^{-1}$. These results show a clear temperature dependance of the particle growth. Indeed, higher temperatures have been shown to favor emission of biogenic precursors, including monoterpenes known to favor the occurrence of NPF events (Kulmala et al., 2004). Previous studies (Paasonen et al., 2018; Yli-Juuti et al., 2011) have shown that the GR usually has larger values during warm periods and especially during summer. Over urban areas (Beijing or Shangai), GR$_{15-25nm}$ showed no clear seasonal variation (Yao et al., 2018). However, recent studies also have highlighted the link with GR seasonal pattern and high abundance of biogenic volatile organic compounds during warmer periods over boreal forest (Paasonen et al., 2018; Yli-Juuti et al., 2011). Therefore, the observed seasonal variation of GR values may be related to temperature variation that influences the emissions of organic compounds in the atmosphere (Figure 5).

As previously observed in Figure 3a, the mean geometric diameters reached by the end of all NPF events are similar and averaged around 50 nm. This can be explained by the presence of a pre-existing mode of particles centered around 50 nm. Moreover, the seasonal variation of the NPF event duration could be then linked to the GR$_{15.7-30nm}$ seasonal variation. As the final diameter is similar in all cases, the lower the GR$_{15.7-30nm}$ values will then be associated with the longer NPF duration. The seasonal variation of NPF duration highlighted earlier is then only a consequence of the GR seasonal variation.

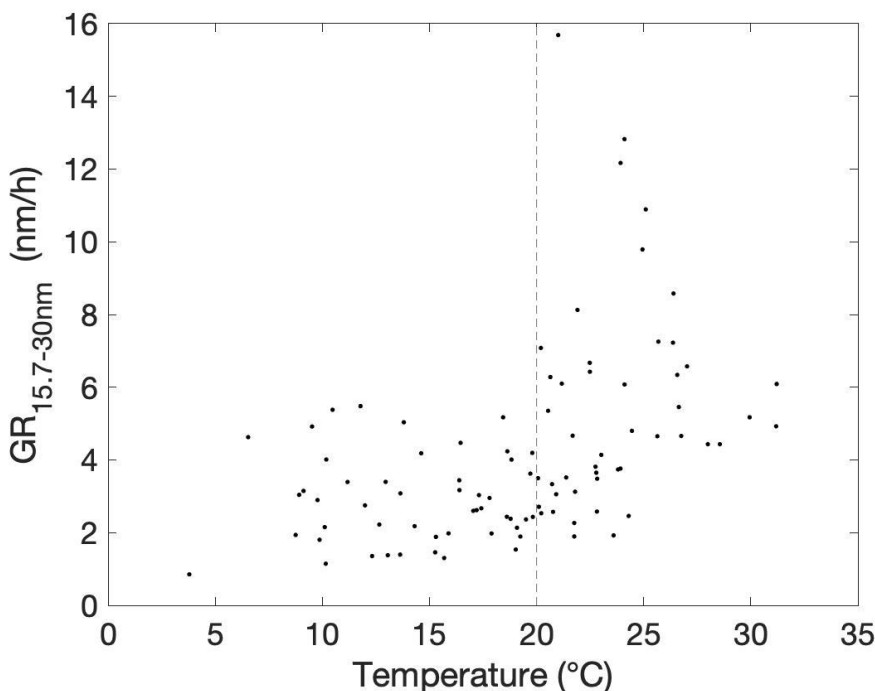

**Figure 5 : Growth rate values (GR$_{15.7-30nm}$ ) as a function of ambient temperature.**

### 3.4 Environmental conditions

The effect of cloudiness on NPF event occurrence is shown in Figure 6a, with a specific focus on
measurements collected between 09:00 and 14:00, i.e. the period of time where most NPF tended to start.
The cloud fraction was calculated from the sky imager dataset following the method by (Shukla et al.,
2016) and sorted as a function of event, undefined and non-event days. There is a clear inverse correlation
between cloud fraction and NPF occurrences. Average cloud fraction is around 0.47 during event days,
0.68 during undefined days and 0.74 during non-event days. Moreover, the 25[th] percentiles of the cloud
fractions for event, undefined and non-event days, respectively 0.06, 0.47, 0.63, clearly show that the
absence of clouds (lower cloud fraction) is mostly associated with NPF events. This result is consistent
with previous analysis performed over the boreal forest (Dada et al., 2017) and is linked to the fact that
radiation seems essential for NPF during the warmer period, as the events occur almost solely during



daylight hours (Kulmala et al., 2004). Figure 6b shows the average diel total solar radiations observed
during events, non-event and undefined days for spring and summer. As expected, total solar radiation is
on average always larger during event days in comparison to non-event days, with a more pronounced
difference observed during spring.

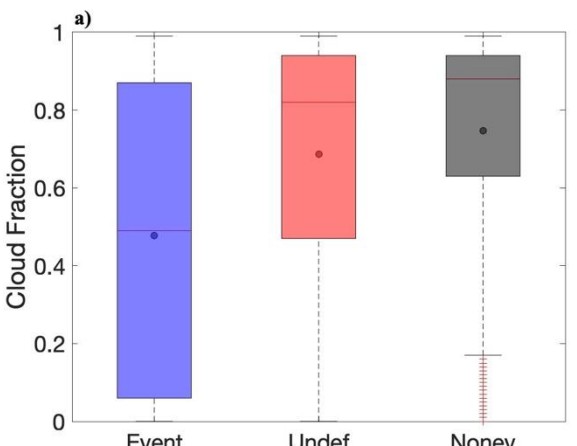
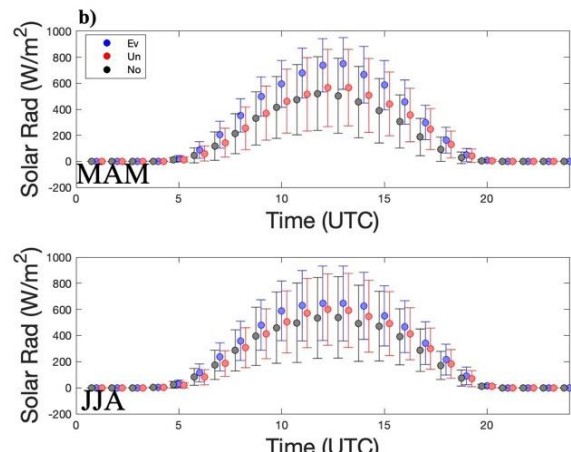

**Figure 6 : (a) Cloud fraction observed from 09:00 to 14:00 UTC during event, undefined and non-event days. The red line represents the median while the lower and upper edges of the box represent the 25th and 75th percentiles, respectively. The lower and upper edges of the whisker represent 10th and 90th percentiles, respectively. The circles represent the average. (b) Diel variations (UTC) of the mean total solar radiation observed during the event days (blue), undefined days (green) and non-event days (red) during spring (MAM, top) and summer (JJA, bottom) seasons (b). The error bars correspond to one standard deviation.**

Other environmental parameters known to influence the occurrence of NPF events, such as temperature
and humidity were also sorted to highlight diel and seasonal variations (Figure 7). Our results (Figure 7a)
indicate that NPF is favored by low values of ambient relative humidity, especially during spring,
consistently with previous studies(Duplissy et al., 2016; Hamed et al., 2011; Merikanto et al., 2016). A
few reasons can explain this tendency: (1) high RH values (RH > 90%) observed at the surface are usually
associated to the presence of low altitude clouds reducing incoming total radiation and then preventing
NPF formation, (2) at moderately high RH (RH >40%), hydrophilic aerosols could growth which will
enlarge the sink for precursors and (3) high RH values limit some VOC (Volatile Organic Compounds)





ozonolysis reactions, which further prevents the formation of condensable vapors necessary for nucleation
(Boy and Kulmala, 2002).
Figure 7b shows the diel median temperature conditions (T) during NPF events, nonevents and undefined
days. NPF events occurred within temperatures ranging between 3°C and 33.5°C. During both seasons,
averaged temperatures during event days are always larger than during non-event days, with, again larger
differences during spring. One should note that days with high temperatures in spring and summer are
usually also days with high solar radiation, consistently with conclusions from Figure 6. The temperature
difference between undefined days and event days is clearly marked during spring and fade away during
summer. As previously discussed, higher temperatures favor emission of biogenic precursors, including
monoterpenes known to favor the occurrence of NPF event (Kulmala et al., 2004). However, high
temperature can also lead to evaporation of molecular clusters which may inhibit NPF events (Dada et
al., 2017; Deng et al., 2020).

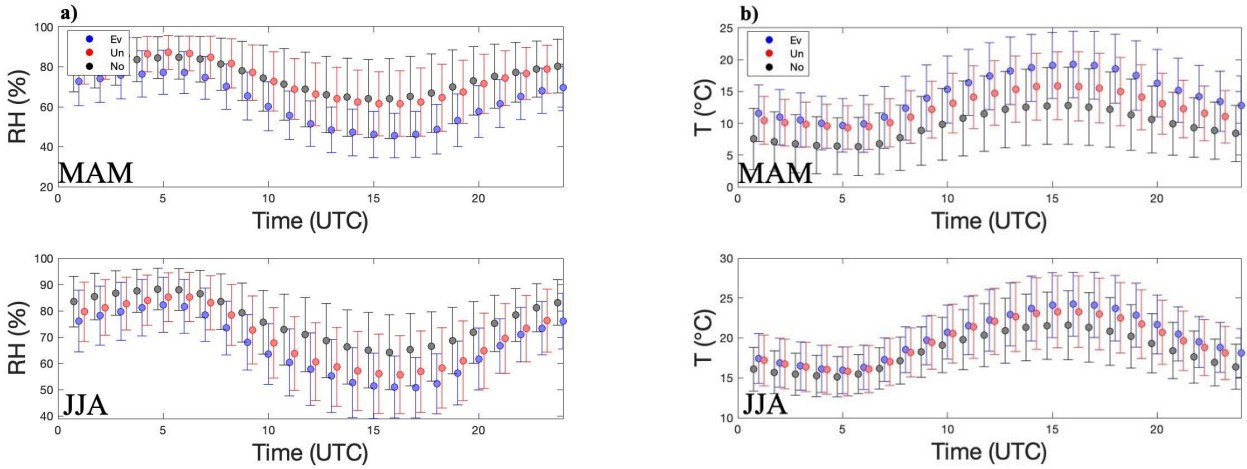

**Figure 7 : Diel variation (UTC) of mean Relative Humidity (RH, a) and mean temperature (b) observed during the event days (blue), undefined days (green) and non-event days (red) during spring (MAM) and summer (JJA) seasons. The error bars correspond to one standard deviation.**



**3.5    Condensation sink**
The CS characterizes the loss rate of atmospheric vapors to aerosol particles. The diel variations of CS
calculated for spring and summer and for NPF event, undefined and non-event days are shown in Figure
8a. Averaged CS values are high (larger than $2 \times 10^{-2}$ s$^{-1}$) during event days occurring during spring and
summer (Figure 8a). During NPF event days and over different urban sites (Beijing, Nanjing or Hong
Kong), CS values ranging from 0.6 up to $10.7 \times 10^{-2}$ s$^{-1}$ were reported (Xiao et al., 2015). Over pristine sites,
such as Hyytiälä, the CS values are between $0.05 – 0.35 \times 10^{-2}$ s$^{-1}$.  As events occur anyway, low values of
CS, often considered as the major limiting factor in the NPF occurrence do not inhibit the occurrence of
NPF events in ATOLL consistently to previous observations in similar environments, such as Melpitz
observatory (Größ et al., 2018) or over Chinese megacities (Xiao et al., 2015). One can assume that the
presence of large concentrations of precursors could explain the formation of particles over polluted sites
such as ATOLL. Unfortunately, precursors were not measured over the 4-year period of interest here
therefore this assumption would require further investigation beyond the scope of this study.
In the afternoon, CS during event days increases due to the growth of freshly emitted particles, especially
during summer. Contribution of newly formed particles (Dp < 50 nm) to the CS is about 36% and 27%,
during summer and spring respectively, while the contribution of pre-existing particles (Dp > 150 nm) to
the CS is below 20% for both seasons. Moreover, during non-event days, the size resolved median CS is
shifted to larger particle diameters with a maximum observed around 100 nm for all seasons.
To evaluate the impact of the background CS on NPF occurrence, all CS values observed from 07:00-
09:00, period before NPF starting time (green area on Figure 4a), were averaged during event, non-event
and undefined days. It was found that the total $CS_{07-09h}$ was larger (around 16%) during non-event days
in comparison to undefined and event days. Moreover, this difference is mostly due to particles larger
than 70 nm according to size resolved $CS_{07-09h}$ (Figure 8b). The difference between non-event and event
days is lower than what is usually observed over pristine sites (Lyubovtseva et al., 2005) but significant
enough to trigger the NPF event occurrence.





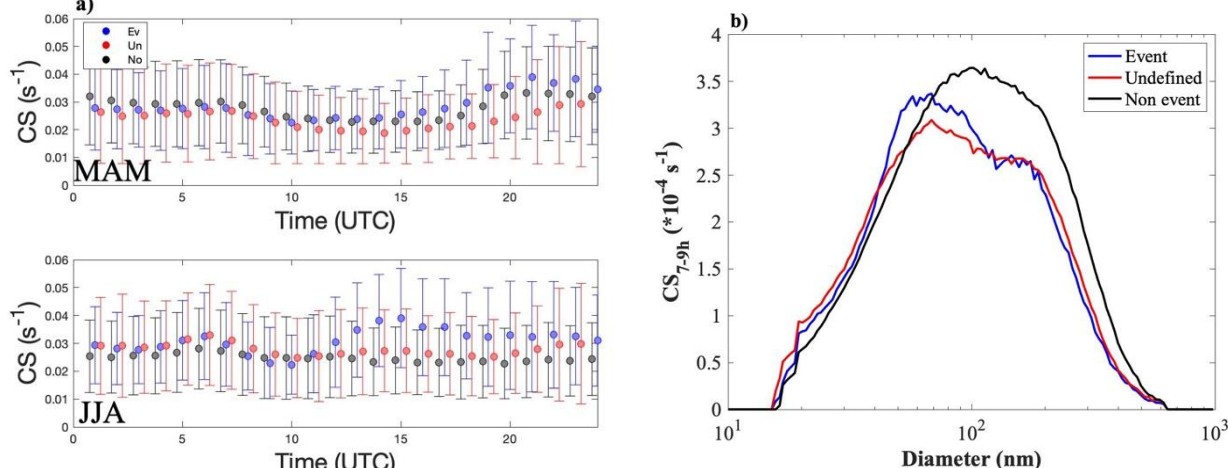

**Figure 8 : (a) Diel variation of Condensation Sink (CS) during spring (MAM) and summer (JJA) seasons. (b) Median size resolved CS for MAM and JJA during event days (blue), undefined (red) and non-event days (black).**

Additionally, the correlation coefficients between meteorological parameters and pollutants (gas and particles) are reported in Table 1 for the entire period of measurements (all seasons). Hourly average over a time window between 09:00 - 14:00 (NPF event starting time period) of few variables (total CS, T, RH and $BC_{wb}$) were used to calculate those correlation coefficients (corresponding to 7025 and 35433 data points for NPF event and Non-event days, respectively).

The correlation of Black Carbon from wood burning ($BC_{wb}$) during non-event days with the condensation sink is high (R = 0.67). This correlation between these parameters is clearly absent during event days (R = 0.19). One can also note that NOx concentrations have a positive correlation (0.30) with CS during NPF non-event days while the same correlation is negative (-0.17) during NPF event days. The NOx sources over urban area are mostly anthropogenic (house heating, traffic and industries) sources which is consistent with its relatively high correlation coefficients with $BC_{wb}$ (0.47 and 0.65). As highlighted in (Barreira et al., 2020), $BC_{wb}$ and NOx are evolving through the year showing a minimum in summer and a maximum in winter when sources are stronger due to colder temperatures and residential heating




emissions. As non-event days are mostly (62%) observed during cold months and NPF events are largely
(82%) observed during warmer months, the correlation between $BC_{wb}$, NOx and CS during non-event is
not surprising. However, during spring, air masses observed during NPF events are clearly "cleaner" (in
terms of NOx and $BC_{wb}$) than non-event cases. Indeed, $NO_x$ and $BC_{wb}$ concentrations are lowered by 18%
and 36% respectively during spring NPF event days in comparison to non-event days. During summer,
$NO_x$ and $BC_{wb}$ concentrations reach an annual minimum and there both pollutant concentrations are
similar between NPF event and non-event days (lowered by -0.04% and 0.01% during NPF event days).
**Table 1 : Correlation coefficients between different meteorological parameters (T, RH), Nitrogen oxide (NOx), Black carbon**
**concentrations (BCwb from wood burning) and total condensation sink during event and non-event for the 4 years period (2017-**
**2020) and in a time window (09:00 – 14:00). High positive or negative correlations are marked in bold.**

|  |  | CS | T | RH | NOx | BCwb |
|---|---|---|---|---|---|---|
| Event days | CS | 1 |  |  |  |  |
|  | T | **0.55** | 1 |  |  |  |
|  | RH | -0.39 | -0.40 | 1 |  |  |
|  | NOx | -0.17 | -0.24 | 0.48 | 1 |  |
|  | BC$_{wb}$ | 0.19 | -0.04 | 0.11 | 0.47 | 1 |
| Non- event days | CS | 1 |  |  |  |  |
|  | T | 0.06 | 1 |  |  |  |
|  | RH | -0.03 | -0.50 | 1 |  |  |
|  | NOx | 0.30 | -0.44 | 0.44 | 1 |  |
|  | BC$_{wb}$ | **0.67** | -0.37 | 0.28 | **0.65** | 1 |

Moreover, during event days the temperature is positively correlated (0.55) with the CS, while, during
non-event days, this correlation is clearly not observed during non-event days (0.06). Over boreal forest,



CS and temperature are correlated during event day (Liao et al., 2014). Indeed, this coupling comes from
the enhanced growth of particles due larger monoterpene emissions at higher temperature, which naturally
leads to higher concentration of larger particles and thus higher CS. As the particle growth during event
days is clearly related to temperature increase (Figure 5) most probably due to higher concentration of
condensable gasses, it is not surprising to observe this temperature and CS coupling.
**3.6    Air mass trajectories**
One can note that environmental conditions (CS, Temperature and RH) observed during undefined events
are mostly between event and non-event days. A deeper analysis on undefined days reveals that on these
days, particle growth stopped due to (i) a decrease of the total irradiance due to a cloud passage over the
site (20% of cases), (ii) a shift of the wind direction (17% of cases), (iii) or both parameters changing
simultaneously (35% of cases).

The shift of the wind orientation leading to a stop of the particle growth indicates that NPF events are
associated with certain wind directions or air mass origins. To investigate this, HYSPLIT back trajectories
were first sorted as a function of event, non-event and undefined days. Only the back-trajectories arriving
between 09:00-14:00 (period of NPF high occurrences) were selected for further analysis. During the
NPF events, the predominant air masses were tracked back along the Eastern North Sea region.
Comparing these results to back trajectories during non-event days highlight more continental influence.
Indeed, most of the back trajectories during non-event days pass over large cities (Dunkirk, Paris, London,
Rotterdam) before reaching Lille metropolis. Those air masses might then have been slightly enriched in
pre-existing particles larger than 100 nm ($CS_{7\_9h}$ slightly larger (16%) during non-event days) which
would decrease the occurrence of NPF events in Lille or could have been depleted in precursor vapors.
This result is consistent with previous results showing "cleaner" air masses are associated with NPF event
cases observed during spring.

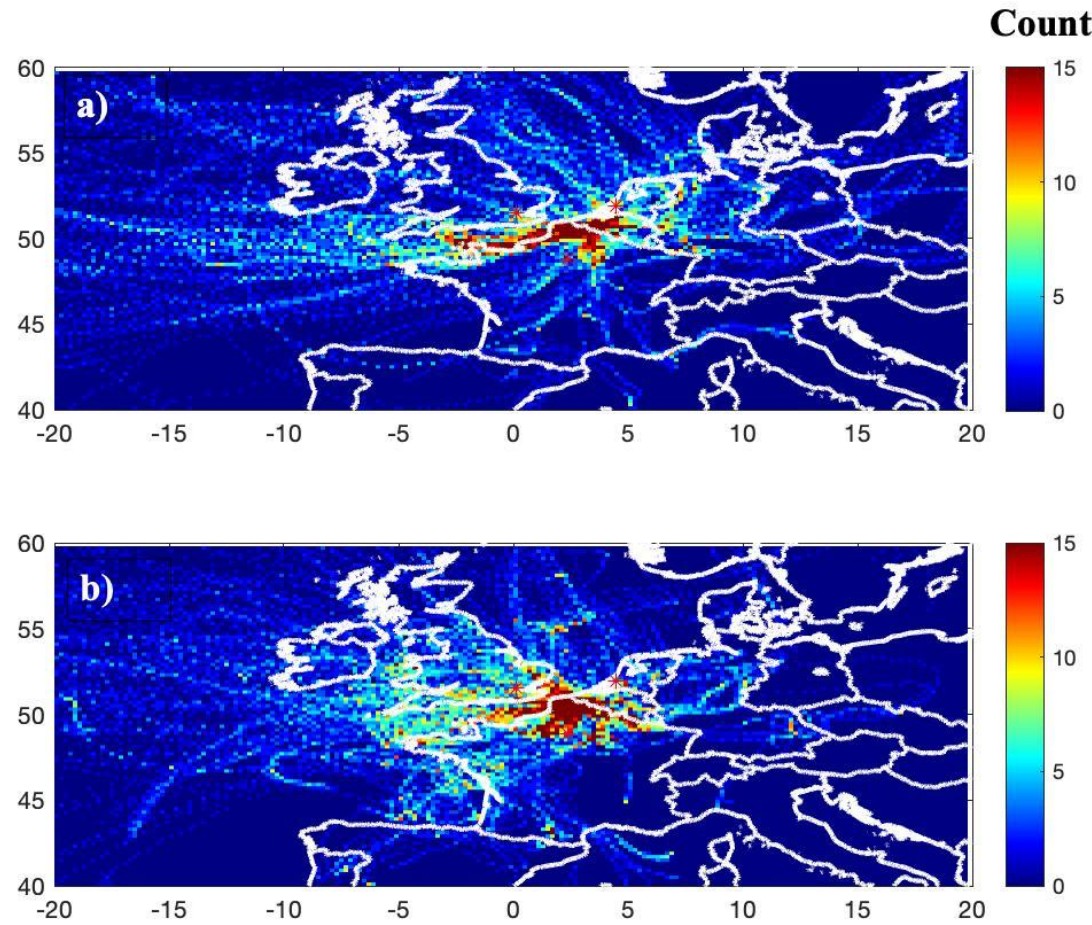

**Figure 9 : 3 days hourly back trajectories arriving in ATOLL between 09:00-14:00 UTC during (a) new particle formation (NPF) events and (b) non-events days. The back trajectories were calculated for each hour at ATOLL at half the boundary layer height. The color contour represents the back trajectories crossing counts in each grid cell (resolution 0.2° · 0.2°).**


### 3.7 Nucleation strength factor

The nucleation strength factor ($NSF_{15.7-100}$) is calculated as the ratio of fine to accumulation particle concentrations observed during nucleation day over the same ratio observed during non-event day (Salma et al., 2017). Fine and accumulation mode particle number concentrations ($N_{15.7-100}$ and $N_{100-800}$) were


retrieved from the SMPS data. The limited atmospheric residence time of fine particles (typically lower
than 10h) means that a large portion of the $N_{15.7-100}$ concentration can also be related to local emissions
and/or formation processes, including NPF events. On the contrary, due to a longer residence time within
the atmosphere (up to 10 days), $N_{100-800}$ is more related to large spatial and temporal scales. Therefore,
the numerator represents the increase of $N_{<100}$ relative to $N_{100-800}$ caused by all sources while the
denominator represents the same property due to all sources except NPF. The NSF method is based on
the hypothesis that aerosol sources are similar from day to day and from season to season, excepting the
sporadic occurrence of NPF. Considering the large number of event (96) and non-event (432) days used
to calculate $NSF_{15.7-100}$, one can assume that the sporadic/occasional (i.e. not observed on daily basis)
sources of UF particles other than NPF events (e.g. volcanic plumes) have little impact on the $NSF_{15.7-100}$
in comparison to the sources always active (such as traffic, industries etc…).
NSF is generally used to better assess the contribution of NPF to fine particle number concentrations
(represented by $N<100$) relative to the regional background particle number concentrations.  If the NSF
$\approx 1$, then the relative contribution of NPF to particle number concentration with respect to other sources
is negligible, like in Granada (Spain) urban site (Casquero-Vera et al., 2021). Moreover, Salma et al.
(2017) also defined two thresholds for NSF6-100 to describe NPF contribution as a single source: a
considerable contribution ($1 < NSF_{6-100} < 2$) or larger than of any other source sectors together ($NSF_{6-100}$
$>2$). One should keep in mind that these thresholds were defined accordingly to the lower cut off diameter
originally set at 6nm. As the lower cut off diameter used in this study is a bit larger (15.7 nm instead of
6nm) than the one used by Salma et al. (2017), the calculated $NSF_{15.7-100}$ would necessarily be
underestimated in comparison to $NSF_{6-100}$ from Salma et al. (2017). The hourly median of fine to
accumulation particle concentration ratio was computed for NPF event and non-event days. Figure 10
shows the $NSF_{15.7-100}$ diel variation observed at the ATOLL platform over 4 years of measurements.
During spring, the $NSF_{15.7-100}$ factor remains quite constant (about 1.5) during night and morning and
peaks at 16:00 UTC to reach a maximum at 2.5. This indicates that NPF has a significant effect on particle
number concentration only a few (2-3) hours after the averaged NPF starting time. During summer, the
tendency of the $NSF_{15.7-100}$ is quite similar with a unique peak at 13:00 UTC (again 2-3 hours after the





averaged NPF starting time). At that time the median $NSF_{15.7-100}$ values reach 4 while from 21:00 to 06:00
UTC the $NSF_{15.7-100}$ remains low (averaged at 1.08). Therefore, during summer, the NPF contribution to
particle number concentration is extremely high from 10:00 to 18:00 and then negligeable for the rest of
the day in comparison to other sources.
Such $NSF_{10-100}$ diel variations were observed in other European cities (Budapest, Vienna and Prague) with
maximum reaching 2.7, 2.3 and 3.4 respectively with a lower cut-off diameter set at 10nm (Németh et al.,
2018). Moreover, Salma et al. (2017) reported $NSF_{6-100}$ peaks at midday varying from 2.2 and 2.7 for
Budapest city center and from 2 to 7.2 for near city background for each season with $NSF_{6-100}$ maximum
reached during winter. The nucleation frequency during winter in Budapest is low (<10%), similarly to
our observations, however, the impact of these limited number of events on particle number
concentrations is high.  For the record, the $NSF_{15.7-100}$ factor peaked at 3.5 and 2.3 during winter and fall
respectively.


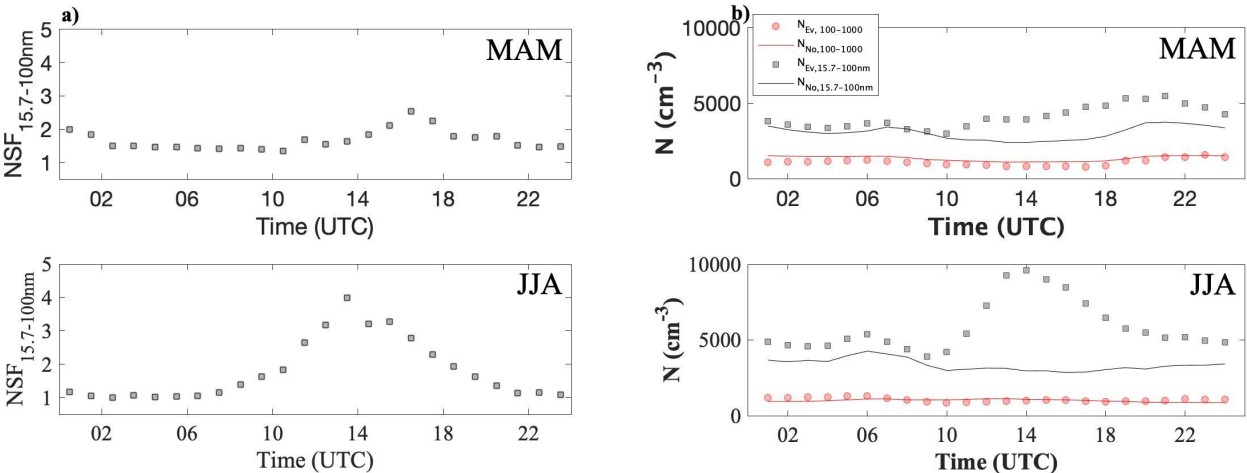

**Figure 10 : (a) Diel variation of the Nucleation Strength Factor ($NSF_{15.7-100}$) during MAM and JJA calculated from number concentration during the 2017-2020 period. (b) Diel variation of particle number concentrations (N) for each season within the diameter ranges from 15.7 to 100 nm ($N_{15.7-100}$, black) and from 100 to 1000 nm ($N_{100-1000}$, red) at the ATOLL site during the 2017-2020 period. The dots correspond to event days while the line correspond to non-event days.**



## 4     Conclusions

This study was based on 4-years (2017- 2020) measurements performed at the ATOLL site, in the close vicinity of the city of Lille, Northern France. This paper is dedicated to studying new particle formation (NPF) occurrence over a peri-urban site. The results highlight a strong seasonal variation of the NPFevent frequency, with a maximum occurrence observed during spring (23 %) and summer (26 %). The undefined cases, which correspond to bursts of UFP that do not grow, are much more frequent (38% on average) than NPF events all year long. The highest frequency (68%) is observed in August and the lowest one (17%) in February. The interruption of the particle growth during undefined events can be  mostly attributed to changes of environmental conditions (irradiance and wind direction).

Seasonal variation of NPF parameters was also clearly observed and associated with environmental parameters. High temperature (T > 295K), low RH (RH< 45%) and high solar radiation favor the occurrence of NPF events at ATOLL. The presence of clouds, linked to a decrease of solar radiation, is limiting the NPF event occurrences. Moreover, NPF events start earlier in the morning during warmer months (May-September) most probably related to variations in sunrise time. The growth rate calculated between 15.7 and 30 nm ($GR_{15.7-30nm}$) ranges from 1.8 nm.h-1 in March up to 10.9 nm.h-1 in July. The $GR_{15.7-30nm}$ was also found to be positively correlated with temperature. This correlation might be related to larger emissions of biogenic precursors at higher temperatures, including monoterpenes known to favor the occurrence of NPF event (Kulmala et al., 2004).

Relatively high values of Condensation Sink (averaged CS > $2.10^{-2}$ $s^{-1}$) are reported during NPF events as well as during non-event days. These results suggest that high CS values are not limiting the NPF event occurrence, consistent with recent studies focusing on NPF events over urban sites (Deng et al., 2020; Hussein et al., 2020; Pushpawela et al., 2018). Looking more closely before the NPF onset (from 07:00 – 09:00 UTC), $CS_{07-09h}$ values are larger by 16% during non-event days. Interestingly, CS tends to increase during event days (especially in summer) and size resolved CS clearly shows a peak shift from 150 nm during non-event days to 50 nm during event days highlighting the strong contribution of newly formed particles on CS.



Air masses trajectories (HYSPLIT) arriving over ATOLL during event days highlight a specific path along the Eastern North Sea region with only a small fraction passing over any continental area and therefore not crossing many anthropogenic sources, while, most of the back trajectories during non-event days pass over large cities (Dunkirk, Paris, London, Rotterdam) before reaching Lille. The precursor vapor concentration and probably their nature might differ from both "clean" and "polluted" air masses and therefore promote or inhibit NPF event occurrences, a point which requires further investigation.

The impact of NPF events on particle number concentrations has been estimated through the nucleation strength factor (NSF; Salma et al., 2017). The $NSF_{15.7-100nm}$ diel variation was calculated for spring and summer occurring 2 to 3 hours after the average NPF starting time and reaching 1.5 and 4 during spring and summer respectively. The extremely large $NSF_{15.7-100nm}$ value observed during summer highlights the very high NPF contribution to the fine particles (Dp < 100 nm) number concentration in comparison to other regional sources. Recently, (Ren et al., 2021) highlighted the strong impact of newly formed particles from NPF on Cloud Condensation Nuclei (CCN) especially at sites close to anthropogenic sources, such as ATOLL. In future studies, the impact of local vertical dynamics such as the effect of boundary layer dynamics as in Lampilahti et al. (2020 and 2021) as well as the CCN enhancement factor will be analysed.

**Acknowledgements**

This research was supported by the French national research agency (ANR) under the MABCaM (ANR-16-CE04-0009) contract. Part of the instrumental system has been financially supported by the CaPPA project (Chemical and Physical Properties of the Atmosphere), which is funded by the French National Research Agency (ANR) through the PIA (Programme d'Investissement d'Avenir) under contract "ANR-11-LABX-0005-01", and by the Regional Council "Hauts-de-France". The authors also thank the Région Hauts-de-France, and the Ministère de l'Enseignement Supérieur et de la Recherche (CPER Climibio), and the European Fund for Regional Economic Development for their financial support. The authors gratefully acknowledge the NOAA Air Resources Laboratory (ARL) for the provision of the HYSPLIT transport and dispersion model and/or READY website (https://www.ready.noaa.gov) used in this publication. We thank Francois Thieuleux for ECMWF data sharing during this work.





# **Data availability**

ATOLL measurements are available through the EBAS database (https://ebas.nilu.no) and SMPS data
before 2020 through `https://doi.org/10.5281/zenodo.6794562`. GDAS files for back-
trajectory calculation are available at https://www.arl.noaa.gov/hysplit/hysplit/ . NOx data are
available from the ATMO open data website : https://data-atmo-hdf.opendata.arcgis.com.

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
