# Peer review of "Measurement report: Atmospheric new particle formation in a peri- # 2 urban site in Lille, Northern France"

_Atmospheric Chemistry and Physics, 2022_

## Referee Comment (RC1)

Referee comment on "Measurement report: Atmospheric new particle formation in a peri-urban site in Lille, Northern France" by Suzanne Crumeyrolle., et al, Atmos. Chem. Phys. Discuss., https://doi.org/10.5194/acp-2022-436, 2022

Anonymous Referee.

Crumeyrolle and co-authors present a measurement report on a long-term dataset from Lille, a large city located in France. The authors performed an extensive analysis of the particle number size distribution and meteorological parameters, in order to explain the factors involved in the new particle formation events over Lille. Crumeyrolle et al., reported that the majority of the NFP observations occurred during spring and summer. It was found that $T > 275K$, $RH < 45$ %, and high solar radiation favored the NPF appearance. Additionally, the authors stated that despite the relatively high CS, new particle formation is observed. This observation is in concordance with other studies in large cities.

I very much appreciate the analysis of this long-term dataset. The results are clearly presented and extensively discussed. The manuscript is a valuable contribution to the field. I would recommend it to be published on ACP after addressing the following comments.

**Specific comments:**

Line 70: what do you mean by promoting growth rate? Do you mean, by promoting an increase in the growth rate?

Line 99: please define GAW.

Line 128: Please rephase "SMPS dry (using a Nafion) particle number size distributions were also used for CS and growth rate (GR)", you can delete growth rate and only use GR, since growth rate was defined before.

Line 192: can you please comment more about this "This might show that ATOLL is under the influence of air masses or particle and precursor sinks that favor the burst of UFP". It is an interesting observation.

Line 219: When the starting time and growth rate are discussed, the authors refer to the local time, not UTC time as in Fig. 3. This certainly is helpful to take into account the dynamic of the site where the monitoring was done. Can you please, refer to the caption of Fig. 4 that the time there is local?

Line 222: please define "colder period". Is it colder than July? I think it is missing the comparative sentence. By looking at Fig. 4, to me, the "colder period" comprises March, April, and May. Or does it include winter?

Line 231: here it is stated that the colder months are (March and November). I think the colder period should be more clearly defined otherwise it can be confusing.

Page 13: Can you please rephrase this sentence "GR15.7-30nm values were, in addition, plotted as a function of temperature for all years and seasons in Figure 5, which highlights that below 20°C, GR15.7-30nm values are lower than 6 nm.h-1, while, under warmer conditions (T >20 °C), GR15.7-30nm reach values up to 16 nm.h-1".

Page 13: can you please explain a bit more why you think this: "As previously observed in Figure 3a, the mean geometric diameters reached by the end of all NPF events are similar and averaged around 50 nm. This can be explained by the presence of a pre-existing mode of particles centered around 50 nm". Thanks

Caption Figure 6. The authors refer to spring (MAM, top) and summer (JJA, bottom) seasons. Is this classification related to the "warmer period"?

Line 259: please rephrase this sentence "at moderately high RH (RH >40%), hydrophilic aerosols could growth which will enlarge the sink for precursors and…"

Line 260: can you please comment briefly on this? "high RH values limit some VOC (Volatile Organic Compounds) ozonolysis reactions, which further prevents the formation of condensable vapors necessary for nucleation". How does the RH affect the ozonolysis reactions?

Line 270: can you please mention that an example of biogenic compounds that inhibit NPF events is isoprene and cite Heinritzi et al., 2020 (Atmos. Chem. Phys., 20, 11809–11821, 2020)?

Can you please give more detail on how the CS was calculated? Please add this information to the method section. Thanks.

In section 3.5, probably https://doi.org/10.1038/s41586-020-2270-4 and Environ. Sci.: Atmos., 2022, 2, 491, can be useful for the discussion.

The author may suggest that monoterpene emissions probably play a role in the observations. Since ATOLL is located at a peri-urban site, are there other precursors possibilities?

For describing Fig. 3, the authors refer to the median, and later (on page 13) they refer to the mean geometric diameter. Can you please comment briefly on how those concepts compare? Are they similar or is there any conversion in between?

Page 13: can you please explain a bit more why you think this: "As previously observed in Figure 3a, the mean geometric diameters reached by the end of all NPF events are similar and averaged around 50 nm. This can be explained by the presence of a pre-existing mode of particles centered around 50 nm". Thanks.

**Technical comments:**

Line 17: please add a dot after "particles" to finalize the sentence.

General: please define how the abbreviations are written, condensation sink (CS) or Condensation Sink (CS)? the same applied to other abbreviations such as UFP, GR, etc.

Line 23: probably it is better to write: using a 4-year long-term dataset, without mentioning the exact date (in the abstract).

Line 29: please change Growth Rate to growth rate.

Line 32: please change "reaches" to "reached" to keep the abstract in past.

Line 35: please change "New Particle Formation" to "New particle formation" to be consistent with the abstract.

Line 36: Please change "The latter" to for example "These particles".

Line 39: Please rephrase this sentence "The freshly-formed particles then grow to larger sizes, from a few nm in particle diameter up to sizes (Dp > 100 nm) at which they may act as cloud condensation nuclei (CCN,..". A possibility could be "The newly-formed particles then grow to larger sizes (Dp > 100 nm) at which they may act as a cloud condensation nuclei (CCN).

Line 52: There is a parenthesis missing at the end of the sentence "Differences were found in both the seasonality and intensity of NPF events according to the site type (urban, traffic, regional, background, rural, polar, high altitude (Dall'Osto et al., 2018; Sellegri et al., 2019)".

Line 124: can you please add which diameter range you consider to be Aitken mode?

Line 67: please change "A recent study (Bousiotis et al., 2021) using large datasets (16 sites) over Europe (6 countries) highlighted…" to "A recent study (Bousiotis et al., 2021) used a large dataset (16 sites) over Europe (6 countries) and highlighted that…"

Line 95: please change "(Métropole Européenne de Lille, more than 1.1 million inhabitants)" to "(Métropole Européenne de Lille with more than 1.1 million inhabitants)"

Line 112: please rephrase "The instruments use in this study focused on aerosol properties including number size distributions, chemical composition, and optical properties, and details are described below". For example, "The instruments used in this study measure the aerosol properties including number, size distributions, chemical composition, and optical properties. The details are described below".

Line 119: please rephrase "Typically, the scan time was chosen to be 300 seconds. To take into account the multiple charge effect and the losses through diffusion, particle concentrations were corrected using the equation given by the manufacturer specifications (AIM 10.2.0.11)" to e.g.,

"The scan time was 300 seconds and the particle concentrations were corrected by taking into account charge effects and diffusion losses".

Line 129: please change "which" to "which".

Line 134: please rephrase "First, the NPF starting time was identified when the newly formed mode was observable in the first bins of the SMPS (15.7 nm) and the time of peak concentrations for particles with a diameter of 30 nm ($N_{30}$) during NPF were manually identified" to "First, the NPF starting time was identified when the newly formed mode was observable in the first bins of the SMPS (15.7 nm). The final time was manually selected and it was defined as the time at which the particle concentration of 30 nm-particles reached a maximum". (For example).

Line 144: please change "5 L.min$^{-1}$" to "5 L min$^{-1}$"

General: sometimes it is written X % and other times X%, please be consistent.

Line 173: please change "(polar, rural, high altitude, remote, urban)" to "(polar, rural, high altitude, remote, and urban)"

Line 205: please change "(See supplementary materials)" to "(see supplementary materials)"

General: There is inconsistency in the font size and font type used along the manuscript, please unify.

Y label of Fig. 4b and along the manuscript: please change "(nm.h$^{-1}$)" to "(nm h$^{-1}$)".

Page 13: please change "Over urban areas (Beijing or Shangai)" to "Over urban areas such as Beijing or Shangai".

Figure 3. Do the plots shown here represent an average or are they representative examples?

Figure 5. please change "(nm/h)" to "(nm h$^{-1}$)".

Figure 6b: please change "(W/m$^2$)" to "(W m$^{-2}$)".

Line 247: please change "total solar radiations" to "total solar radiation".

Line 255: there is one line spacing missing between studies and (Duplissy…).

Line 262: please change temperature conditions (T) to (T).

General: I would recommend increasing the font size of the axis on the plots, for example in Figure 6b, Figure 7, etc.

Line 276: please change "(larger than 2 10-2 s-1)" to "(larger than 2.10-2 s-1)" or "(larger than to 2e-2 s-1)". The same for lines 278 and 279.

Line 359: typically, there is a space between the number and the unit, e.g., 10 h instead of "10h".

---

## Author Comment (AC1)

*We fully acknowledge the comments and suggestions of reviewer 1 and hope to have improved the (revised) manuscript accordingly.*

Crumeyrolle and co-authors present a measurement report on a long-term dataset from Lille, a large city located in France. The authors performed an extensive analysis of the particle number size distribution and meteorological parameters, in order to explain the factors involved in the new particle formation events over Lille. Crumeyrolle et al., reported that the majority of the NFP observations occurred during spring and summer. It was found that T > 275K, RH < 45 %, and high solar radiation favored the NPF appearance. Additionally, the authors stated that despite the relatively high CS, new particle formation is observed. This observation is in concordance with other studies in large cities.

I very much appreciate the analysis of this long-term dataset. The results are clearly presented and extensively discussed. The manuscript is a valuable contribution to the field. I would recommend it to be published on ACP after addressing the following comments.

**Specific comments:**

*Line 70: what do you mean by promoting growth rate? Do you mean, by promoting an increase in the growth rate?*

Initially the sentence was :

"A recent study (Bousiotis et al., 2021) using large datasets (16 sites) over Europe (6 countries) highlighted that solar radiation intensity, temperature, and atmospheric pressure had a positive relationship with the occurrence of NPF events at the majority of sites (exceptions were found for the southern sites), either promoting particle formation or growth rate."

Now it reads:

"A recent study (Bousiotis et al., 2021) using large datasets (16 sites) over Europe (6 countries) highlighted that solar radiation intensity, temperature, and atmospheric pressure had a positive relationship with the occurrence of NPF events at the majority of sites (exceptions were found for the southern sites), either promoting particle formation or increasing growth rate."

Line 99: please define GAW.

GAW definition (Global Atmospheric Watch) has been added to the manuscript

*Line 128: Please rephase "SMPS dry (using a Nafion) particle number size distributions were also used for CS and growth rate (GR)", you can delete growth rate and only use GR, since growth rate was defined before.*

This was done.

*Line 192: can you please comment more about this "This might show that ATOLL is under the influence of air masses or particle and precursor sinks that favor the burst of UFP". It is an interesting observation.*

The comparison between the study performed in Paris, 200 km south of Lille, and our results highlighted that bursts of UFP are observed more often over ATOLL than over Paris. As the paper focuses on NPF events and not burst, the authors have not yet looked in details at those burst events. These kinds of events may be linked to atmospheric dynamics as highlighted by Lampilahti et al. (2020 and 2021) and we plan to look into that since we purchased and installed in 2020 at the site an ultra-sonic anemometer. It could also be related to precursors concentration variations. However, at that time we did not have any instrument monitoring instrument. Since then, we also installed a SO2 analyser. Therefore, in order to fully answer your comments, we would have to work on a more recent data set.

*Line 219: When the starting time and growth rate are discussed, the authors refer to the local time, not UTC time as in Fig. 3. This certainly is helpful to take into account the dynamic of the site where the monitoring was done. Can you please, refer to the caption of Fig. 4 that the time there is local?*

The time in the figure 4 was in fact UTC. Thanks for pointing that mistake in the caption. The reviewer is right to note that local time would make it easier to highlight the local dynamics. However, as most of the events (except two in March) are occurring during summer daylight saving time, the plot would be highly similar (see below) to the one presented in the submitted manuscript. To avoid any misunderstanding, the Figure 4 will remain in UTC time.

[Figure]

**Monthly variation of new particle formation starting time in UTC (a) and in local time (b)**

*Line 222: please define "colder period". Is it colder than July? I think it is missing the comparative sentence. By looking at Fig. 4, to me, the "colder period" comprises March, April, and May. Or does it include winter?*

You are right, the colder period corresponds to March April and May. The results are also true for winter and fall but not shown here due to statistical issue (only few cases of NPF events). We added a sentence to clarify this point.

Previous sentence: "Indeed, the NPF starting time becomes earlier during the colder period and reaches a minimum in June (around 08:20)."

Corrected sentence: "Indeed, the NPF starting time occurs later during spring (also true for fall and winter) and reaches a minimum in June (around 08:20)."

*Line 231: here it is stated that the colder months are (March and November). I think the colder period should be more clearly defined otherwise it can be confusing.*

This sentence has been rephrased to avoid the used of colder months:" On average, NPF duration is shorter from May to August (around 8 hours) and increases up to around 13 hours on average in March and November."

*Page 13: Can you please rephrase this sentence "GR15.7-30nm values were, in addition, plotted as a function of temperature for all years and seasons in Figure 5, which highlights that below 20°C, GR15.7-30nm values are lower than 6 nm.h-1, while, under warmer conditions (T >20 °C), GR15.7-30nm reach values up to 16 nm.h-1".*

The sentence now reads :

"All $GR_{15.7-30nm}$ values were plotted in Figure 5 as a function of temperature. This figure highlights that $GR_{15.7-30nm}$ values are always lower than 6 nm.h$^{-1}$ for low temperatures (T<20°C), while, under warmer conditions (T >20 °C), $GR_{15.7-30nm}$ reach values up to 16 nm h$^{-1}$."

The figure was also modified to highlight the different seasons and highlight the tendency:

[Figure]

**New Figure 5 : Growth Rate (GR$_{15.7-30nm}$ ) values as a function of ambient temperature for different seasons.**

*Page 13: can you please explain a bit more why you think this: "As previously observed in Figure 3a, the mean geometric diameters reached by the end of all NPF events are similar and averaged around 50 nm. This can be explained by the presence of a pre-existing mode of particles centered around 50 nm". Thanks*

The second sentence could suggest that the growth of new particles was stopped at 50 nm because of the presence of pre-existing particles at this same size; this is not the message we wanted to give, so we removed the second sentence to avoid any misunderstanding. Explanations for the growth termination require a more detailed investigation in relation to the atmospheric dynamics (and in particular to the variability of the boundary layer height) and also in relation to the area where the process is initiated / takes place.

The authors plan in a future study to run Nanomap (Kristensson et al., 2014) on the data set to explore the area where the NPF events take place and to obtain statistics of the geographical occurrence of NPF events with dense spatial coverage and precision. NanoMap could also be used for the determination of the horizontal extent of an event combined with observations from site close to ATOLL such as Paris (200 km south) and Brussels (100 km east). This analysis could help us better understand regional events (Class I), in particular whether the growth stops at all sites simultaneously or only in Lille.

However in the absence of strong evidence supporting our assumptions, this sentence was reformulated to avoid any confusion: "As previously observed in Figure 3a, the median diameters reached by the end of all NPF events are similar and averaged around 50 nm. "

*Caption Figure 6. The authors refer to spring (MAM, top) and summer (JJA, bottom) seasons. Is this classification related to the "warmer period"?*

We have checked throughout the manuscript for the occurrence of warmer/colder period wording in order to clarify this. In this case the warmer period (L.257) indeed corresponds to both spring and summer. " This result is consistent with previous analysis performed over the boreal forest (Dada et al., 2017) and is linked to the fact that radiation seems essential for NPF during the warmer period (spring and summer), as the events occur almost solely during daylight hours (Kulmala et al., 2004)."

*Line 259: please rephrase this sentence "at moderately high RH (RH >40%), hydrophilic aerosols could growth which will enlarge the sink for precursors and..."*

Initial sentence was: "at moderately high RH (RH >40%), hydrophilic aerosols could growth which will enlarge the sink for precursors and"

Now reworded as:" at moderately high RH (RH >40%), hydrophilic aerosols could grow which will enlarge the sink for precursors and"

*Line 260: can you please comment briefly on this? "high RH values limit some VOC (Volatile Organic Compounds) ozonolysis reactions, which further prevents the formation of condensable vapors necessary for nucleation". How does the RH affect the ozonolysis reactions?*

More relevant references have been added. According to previous studies (Fick et al., 2003; Tillmann et al., 2010), the presence of RH can change the products of the ozonolysis. For example, Tillmann et al. (2010) have been running different ozonolysis experiments within the AIDA chamber at different temperatures and relative humidities. They found out that at high RH (68%), the α-pinene is consumed and the pinonaldehyde is then formed. Since, this compound is mostly in the gas phase at room or higher temperatures, it won't generate Secondary Organic Aerosols (SOA) by condensation. Therefore, the RH can change the products formed via the ozonolysis reactions and tend to limit the production of SOA.

The sentence was changed to: "high RH values may limit the formation of some Volatile Organic Compounds (VOC) through ozonolysis reactions, inhibiting the formation of condensable vapors necessary for condensation."

*Line 270: can you please mention that an example of biogenic compounds that inhibit NPF events is isoprene and cite Heinritzi et al., 2020 (Atmos. Chem. Phys., 20, 11809–11821, 2020)?*

This is now included : "As previously discussed, higher temperatures favor emission of biogenic precursors, including monoterpenes known to favor the occurrence of NPF event (Kulmala et al., 2004). Isoprene emission is also larger at higher temperature, but according to Heinritzi et al., (2020) its presence can make the difference between measurable new-particle formation events and their absence. Moreover, high temperature can also lead to evaporation of molecular clusters which may inhibit NPF events (Dada et al., 2017; Deng et al., 2020)."

*Can you please give more detail on how the CS was calculated? Please add this information to the method section. Thanks.*

We now included the equations used to calculate the CS :

"SMPS dry (using a Nafion) particle number size distributions were also used for CS ($CS = 2\pi D \sum_i \beta_{Mi} d_{p,i} N_i$      Equation 1, where $\beta_{Mi}$ is the transitional correction factor (Fuchs and Sutugin, 1970), the Knudsen number is $Kn = 2\lambda_v / d_p$, and α is the accommodation coefficient and set to unity here)…"

$$CS = 2\pi D \sum_i \beta_{Mi} d_{p,i} N_i \qquad\qquad\qquad\qquad\qquad \textbf{Equation 1}$$

$$\beta_{Mi} = \frac{1+K_n}{1+0.337Kn+\frac{4}{3}\alpha^{-1}Kn+\frac{4}{3}\alpha^{-1}Kn^2}$$     **Equation 2**

*In section 3.5, probably https://doi.org/10.1038/s41586-020-2270-4 and Environ. Sci.: Atmos., 2022, 2, 491, can be useful for the discussion.*

Thanks for the interesting references that are now added to the manuscript. "Recent studies (Marten et al., 2022; Wang et al., 2020), performed in the CLOUD chamber, demonstrate that the presence of nitric acid ($HNO_3$) and ammonia ($NH_3$), typical within urban environments, contribute to freshly formed particles survival by increasing their growth rate."

*The author may suggest that monoterpene emissions probably play a role in the observations. Since ATOLL is located at a peri-urban site, are there other precursors possibilities?*

Unfortunately, precursors were not measured over the period of interest here, thus the assumptions cannot be tested. The site is located inside a university campus with abundant tree plantings, which is why the monoterpenes were suggested as possible summertime precursors.

*For describing Fig. 3, the authors refer to the median, and later (on page 13) they refer to the mean geometric diameter. Can you please comment briefly on how those concepts compare? Are they similar or is there any conversion in between?*

When statistically describing (fitting) lognormal distributions, the geometric mean diameter of normal distributions is replaced by the count median diameter (CMD). In lognormal distributions, the log of the particle size distribution is symmetrical, so the mean and the median of the lognormal distribution are equal.

The median of the lognormal distribution and normal distribution are equal, since the order of the values does not change when converting to a lognormal distribution. Therefore, for a lognormal distribution, $D_g$ = CMD.

$D_g$ =CMD= $(D_1^{n1} \cdot D_2^{n_2} \cdot D_3^{n_3} \cdots D_N^{n_n})^{1/N}$

where:

$D_g$ = geometric mean diameter

$D_i$ = midpoint particle size

$n_i$ = number of particles in group i having a midpoint size $D_i$

$N = \sum n_i$, the total number of particles, summed over all intervals

In this case, we did not fit the aerosol SD but we used the median diameter, so we removed the term "mean geometric diameter" and replaced it by 'median'.

**Technical comments:**

*Line 17: please add a dot after "particles" to finalize the sentence.*

Done

*General: please define how the abbreviations are written, condensation sink (CS) or Condensation Sink (CS)? the same applied to other abbreviations such as UFP, GR, etc.*

Done

*Line 23: probably it is better to write: using a 4-year long-term dataset, without mentioning the exact date (in the abstract).*

Done

*Line 29: please change Growth Rate to growth rate.*

As we choose to define the abbreviations using capitals, we kept the Growth Rate as is.

*Line 32: please change "reaches" to "reached" to keep the abstract in past.*

Done

*Line 35: please change "New Particle Formation" to "New particle formation" to be consistent with the abstract.*

Again, As we choose to define the abbreviations using capitals, we kept the New Particle Formation as is and only use NPF in the following sections.

*Line 36: Please change "The latter" to for example "These particles".*

Done

*Line 39: Please rephrase this sentence "The freshly-formed particles then grow to larger sizes, from a few nm in particle diameter up to sizes (Dp > 100 nm) at which they may act as cloud condensation nuclei (CCN,..". A possibility could be "The newly-formed particles then grow to larger sizes (Dp > 100 nm) at which they may act as a cloud condensation nuclei (CCN).*

It was corrected as suggested.

*Line 52: There is a parenthesis missing at the end of the sentence "Differences were found in both the seasonality and intensity of NPF events according to the site type (urban, traffic, regional, background, rural, polar, high altitude (Dall'Osto et al., 2018; Sellegri et al., 2019)".*

This was added.

*Line 124: can you please add which diameter range you consider to be Aitken mode?*

The new sentence now reads: 'and their consequent growth to Aitken mode (Dp < 80 nm)'

*Line 67: please change "A recent study (Bousiotis et al., 2021) using large datasets (16 sites) over Europe (6 countries) highlighted..." to "A recent study (Bousiotis et al., 2021) used a large dataset (16 sites) over Europe (6 countries) and highlighted that..."*

Corrected as suggested.

*Line 95: please change "(Métropole Européenne de Lille, more than 1.1 million inhabitants)" to "(Métropole Européenne de Lille with more than 1.1 million inhabitants)"*

Corrected as suggested.

*Line 112: please rephrase "The instruments use in this study focused on aerosol properties including number size distributions, chemical composition, and optical properties, and details are described below". For example, "The instruments used in this study measure the aerosol properties including number, size distributions, chemical composition, and optical properties. The details are described below".*

It was corrected as suggested.

*Line 119: please rephrase "Typically, the scan time was chosen to be 300 seconds. To take into account the multiple charge effect and the losses through diffusion, particle concentrations were corrected using the equation given by the manufacturer specifications (AIM 10.2.0.11)" to e.g.,*

*"The scan time was 300 seconds and the particle concentrations were corrected by taking into account charge effects and diffusion losses".*

This was modified into : "The scan time was 300 seconds and the particle concentrations were corrected by taking into account charge effects and diffusion losses calculated using the manufacturer software and algorithms (AIM 10.2.0.11)."

*Line 129: please change "which" to "which".*

Done.

*Line 134: please rephrase "First, the NPF starting time was identified when the newly formed mode was observable in the first bins of the SMPS (15.7 nm) and the time of peak concentrations for particles with a diameter of 30 nm ($N_{30}$) during NPF were manually identified" to "First, the NPF starting time was identified when the newly formed mode was observable in the first bins of the SMPS (15.7 nm). The final time was manually selected and it was defined as the time at which the particle concentration of 30 nm-particles reached a maximum". (For example).*

It was corrected as suggested.

*Line 144: please change "5 L.min⁻¹" to "5 L min⁻¹"*
*General: sometimes it is written X % and other times X%, please be consistent.*

We modified the flow units and checked the whole manuscript to keep percentage as xx %.

*Line 173: please change "(polar, rural, high altitude, remote, urban)" to "(polar, rural, high altitude, remote, and urban)"*

It was corrected as suggested.

*Line 205: please change "(See supplementary materials)" to "(see supplementary materials)"*

Done.

*General: There is inconsistency in the font size and font type used along the manuscript, please unify.*

We are so sorry for that and we of course modified it in the revised version of the manuscript

*Y label of Fig. 4b and along the manuscript: please change "(nm.h⁻¹)" to "(nm h⁻¹)".*

It was corrected as suggested.

*Page 13: please change "Over urban areas (Beijing or Shangai)" to "Over urban areas such as Beijing or Shangai".*

*Done*

*Figure 3. Do the plots shown here represent an average or are they representative examples?*

To plot Figure 3, we used all the data recorded during spring and summer. We selected all days when a NPF event was observed and then we averaged the data to one-hour time resolution using median filtering as in Kulmala et al (2022). This was clarified within the manuscript. "Median daily contour plots of the particle number size distributions (PNSD) obtained from the SMPS are shown in Figure 3 separately for NPF event, undefined and non-event days observed during the warm period (only spring and summer). All the aerosol size distributions observed during NPF event (around 800 PNSD), undefined (around 2300 PNSD) and non-event (around 1700 PNSD) days were selected then averaged to one-hour time resolution using median filtering."

*Figure 5. please change "(nm/h)" to "(nm h⁻¹)".*
*Figure 6b: please change "(W/m²)" to "(W m⁻²)".*

The units were changed accordingly to the suggestions.

*Line 247: please change "total solar radiations" to "total solar radiation".*

Done

*Line 255: there is one line spacing missing between studies and (Duplissy...). Line 262: please change temperature conditions (T) to (T).*

This was corrected.

*General: I would recommend increasing the font size of the axis on the plots, for example in Figure 6b, Figure 7, etc.*

We did increase the font size on all diel figures. We also added a grid to improve the readings.

*Line 276: please change "(larger than 2 10-2 s-1)" to "(larger than 2.10-2 s-1)" or "(larger than to 2e-2 s-1)". The same for lines 278 and 279.*

This was corrected in $2 \times 10^{-2}$ s$^{-1}$

*Line 359: typically, there is a space between the number and the unit, e.g., 10 h instead of "10h".*

The error was corrected throughout the manuscript.

[revised manuscript text omitted]

---

## Author Comment (AC2)

We would like to thank the reviewer 2 for his comments and his/her suggestions that have improved the quality of the manuscript.

*This work reports long-term (4 years) measurements of particle number size distribution at an urban site in Lille, North of France. This study aims to better understand the environmental factors favoring/disfavoring atmospheric new particle formation in this urban environment. These studies allow to reduce the lack of knowledge that still exist on the process of new particle formation and their subsequent growth. It is a complex and extended dataset and analysis and the results will fit within the scope of ACP, being of interest for the international research community. However, I would suggest some aspects to be considered in order to improve the manuscript and/or strengthen its impact before it is published in ACP.*

Major comments

*The dataset presented is of interest for the international community and combine a large number analysis. However, the manuscript is mainly descriptive, the results of each section are not analyzed/discussed in deep and not big conclusions are reached. I could suggest the authors to include more discussion about GR (maybe the contribution of H2SO4 to GRs and/or improve the discussion of possible precursors -comment below-), include Formation Rate analysis, the differences between event and non event days (in deep analysis) and try to investigate the differences on the CS. In this sense, since 1) the introduction is mainly focus on urban areas and CS effect on NPF events and 2) ATOLL has some measurements of aerosol chemical composition (with ACSM), I would also suggest to look if there is some relationship with the chemical composition of pre-existing particles acting as CS (two recent reads about CS efficiency Du et al. (2022) and Marten et al. (2022)). The ACSM measures from ~80nm and it's a good estimation of the CS chemical composition. This links with the fact that BCwb is high during non-event days, and recent study (Yus-Diez et al., 2022) have shown the impact of secondary aerosol on this quantity.*

Thanks for this comment. At the moment the authors wrote this manuscript those papers were probably not yet published. However, their results are extremely interesting and need to be included into our manuscript.

The Figure 1 is showing the $PM_{2.5}$ as a function of the CS both averaged over the period where most of the NPF are observed (09:00 to 14:00) for NPF event (red) and non-event (grey) days. This figure similar to the one on Du et al. (2022) results is clearly different from the Beijing situation. Indeed, over Beijing the NPF event are clearly associated with lower values of CS and low $PM_{2.5}$ concentrations. Over ATOLL, the NPF events are not clearly associated with low CS values. In fact, NPF events and non-events are occurring over the same range of CS values ($0.03 – 0.7 \ s^{-1}$). However, one can see that the $PM_{2.5}$ concentrations are on average lower during NPF event days in comparison to NPF non-event days.

[Figure]

*Figure 1 : Daytime average (from 09:00 to 14:00) PM2.5 concentration vs daytime average (from 09:00 to 14:00) condensation sink (CS).*

An ACSM is monitoring the aerosol chemical composition on the ATOLL station. Du et al (2022) highlighted the chemical composition of the particles that contribute to the CS. The comparison of the chemical mass fraction observed during NPF event days and non-event days highlight the most effective chemical mass fraction in taking up condensable vapors. They observed a large increase of nitrate and a decrease of organics with the CS values. Moreover, the chemical mass fraction highlights a large increase of Nitrate during non-event days in comparison to NPF event days. As nitrate is highly hygroscopic, nitrate enriched particles are more likely to adsorb water and grow to larger sizes promoting the uptake of gaseous vapors. These observations, consistent with simultaneous kappa calculations, suggest that the particle chemical composition affects the efficiency of the CS for both condensable and reactive uptake of vapor molecules.

In our study, we highlight that the CS is largely influenced by the freshly formed particles, so Figure 2 presents the chemical composition of the particle as a function of CS during two specific periods: before (07:00 – 09:00) and during (09:00 – 14:00) the NPF periods for NPF event and non-event days (Figure 2). During non-event days, both periods (07:00 – 09:00 and 09:00 – 14:00) exhibit similar mass fraction of all compounds with on average 41% Organics, 16% of nitrate, 21% of sulfate, 11% of ammonium, less than 1% of chloride and around 10% of Black Carbon. As the aerosol sources during non-event days are supposed to be the same throughout the day, this result was actually expected.

The same comparison, for event days, shows a larger contribution of Organics to the CS during the NPF period (54% on average) in comparison to the period right before the start of the NPF events (46% in average). Indeed, for large values of CS (> 0.045) the contribution of Organics is larger than 50%, reaching a maximum of 69% for CS of 0.085 $s^{-1}$. This result suggests that organic vapors are likely involved in the particle growth.

[Figure]

*Figure 2 : Mass fraction of the major compound measured before (07:00 – 09:00; left side) and during (09:00 – 14:00; right side) NPF periods at the ATOLL station during event (a and c) and non-event days. The black dashed line corresponds to 50% mass fraction.*

Part of this discussion has been added to the supplementary material as it contributes to providing clues on the involvement of organic compounds in particle growth. However, we believe that a deeper analysis would be required to fully explore the role of the particle chemical composition on CS efficiency, and, similar to the referred papers (Du et al. 2022, Marten et al. 2022), this could be the subject of a dedicated study. Moreover, the relevance of such a detailed study in the present work might be questionable since the CS does not appear as a limiting factor for the occurrence of NPF, which remains the main focus of the paper.

Minor comments
L36 – "highly // significantly" contribute.
This was corrected
L63 – I would use "NPF event" instead of "nucleation event", nucleation is just the process of formation.
Of course, the reviewer is right and this has been corrected as suggested.
L71 – I would include reference (Dada et al. 2007 fits well).
The reference was included to support the link between high temperatures and the growth of newly formed particles
L94 – Indicate altitude
The altitude of the station has been added.
L97 – This reference is not included in the reference list. Include at least title?
The reference has been added.
Velazquez-Garcia[1,2]*A., S. Crumeyrolle[2]*, J.F. de Brito[1], E. Tison[1], E. Bourrianne[2], I. Chiapello[2], V. Riffault[1]. Deriving composition-dependent aerosol absorption, scattering and extinction mass efficiencies from multi-annual high time resolution observations in Northern France. Submitted to Atmospheric Environment, Apr. 2022.

L99 – Rose et al. 2021 maybe fits better?
The authors choose to keep Laj et al. and add Rose et al.
L104 – volcanic plumes affect the surface levels?
Yes, the authors added a reference (Boichu et al. 2019) in which they explore month/year long ACSM data in France from stations in contrasting environments. They have shown that volcanic

sulfate aerosols exhibit a distinct chemical signature in urban/rural conditions, with $NO_3$:$SO_4$ mass concentration ratios lower than for non-volcanic background aerosols.

You are right, the SMPS is a 3082 but the DMA is a 3081A. This sentence has been corrected.

The sentence is now : "The classification procedure, presented in Dal Maso et al. (2005), is following the decision criteria based on the presence of fine particles (Dp < 25 nm) and their consequent growth to Aitken mode (Dp < 80 nm)."

Reviewer 1 suggested to include the equations used to describe the CS calculations. So we included both transitional correction factor ($\beta$  ) and CS equations into the revised version of the manuscript. The Knudsen number is defined as the ratio of the molecular mean free path length to a representative physical length scale. The mean free path here was then calculated assuming all condensable vapors were sulfuric acid. Therefore, we used the molar mass and the diffusion volume of sulfuric acid. For those calculations, the chemical composition of the preexisting particles is not considered. However, as the reviewer 2 suggested, we added a part to describe the CS chemical composition and it's influence on the condensation efficiency.

Right, the sentence was corrected accordingly. We meant to say that the more preexisting particles, the larger the probability for clusters to coagulate on it.

The whole paragraph was modified into :
"The particle $GR_{15.7 - 30}$, from 15.7 to 30nm, was calculated based on the maximum-concentration method described in (Kulmala et al., 2012). First, the NPF starting time was identified when the newly formed mode was observable in the first bin of the SMPS (15.7 nm). Then, the time when the concentrations for particles with diameter of 30 nm ($N_{30}$) peaked was alsomanually identified. Particle $GR_{15.7 - 30}$ was then calculated by linear regression of particle size vs. time span from the NPF start until time when $N_{30}$ reaches a maximum ($GR = (D_{p,2} - D_{p1})/(T_2 - T_1)$)"

The

The

"SMPS dry (using a Nafion) particle number size distributions were also used for CS ($CS = 2\pi D_i \beta_{Mi} d_{p,i} N_i$                                        Equation 1, where $\beta_{Mi}$ is the transitional correction factor (Fuks and Sutugin, 1970), the Knudsen number is $Kn = 2\lambda_v/d_p$, and $\alpha$ is the accommodation coefficient and set to unity here) and GR calculations. CS estimates the loss rate of the condensable vapors (Kulmala et al., 2001) which were assumed to have molecular properties similar to sulfuric acid (Dal Maso et al., 2005). A high CS indicates the presence of large surface area of aerosol particles onto which NPF precursors can condensate. The particle $GR_{15.7 - 30}$, from 15.7 to 30nm, was calculated based on the maximum-concentration method described in (Kulmala et al., 2012). First, the NPF starting time was

concentration method described in (Kulmala et al., 2012). First, the NPF starting time was identified when the newly formed mode was observable in the first bin of the SMPS (15.7 nm). Then, the time when the concentrations for particles with diameter of 30 nm ($N_{30}$) peaked was alsomanually identified. Particle $GR_{15.7 - 30}$ was then calculated by linear regression of particle size vs. time span from the NPF start until time when $N_{30}$ reaches a maximum ($GR = D_{p,2} - D_{p1}/T_2 - T_1$)

$$CS = 2\pi D \sum_i \beta_{Mi} d_{p,i} N_i \qquad\qquad\qquad \textit{Equation 1}$$

$$\beta_{Mi} = \frac{1+K_n}{1+0.337Kn+\frac{4}{3}\alpha^{-1}Kn+\frac{4}{3}\alpha^{-1}Kn^2} \qquad\qquad \textit{Equation 2}$$

144,145 – "L.min" change by "L·min". "Wood Burning" not necessarily capital letters. This was corrected

L148, 149 – indicate what is each instrument measuring and how you use it. About the solar radiation, only global seems to appear in the results sections.
The solar irradiance on the horizontal plane is measured every minute with two instruments, a pyrheliometer and a pyranometer. Both are associated with an EKO automated sun tracking system and both are Kipp&Zonen instruments (CHP1 and CMP22) with a close spectral range and response (around 300 nm to 3600 nm at half maximum). The pyrheliometer provides the direct normal irradiance (DNI) and the pyranometer measures the diffuse horizontal solar irradiance (DifHI) using a sun shading ball. The global horizontal solar irradiance (GHI) is obtained as follows: GHI = DNI x cos(SolarZenitalAngle) + DifHI. The quality of the instruments and the automated shading device ensures the accuracy of the measurements to around 1% for GHI under clear sun conditions, and several percents for GHI and DifHI depending on the magnitude of signal (and the presence of clouds).

The sky imager is a camera equipped with a fisheye lens to cover the entire upper half sphere. The cloud cover is estimated from an algorithm comparing the different values of the red, green and blue components of each pixel of the taken images.

Part of this discussion was added to the manuscript :
"Solar radiation at the surface are measured every minute at the sampling site using a set of Kipp & Zonen pyranometer (CM22, for diffuse fluxes using a sphere shadower) and Normal Incidence pyrheliometer (CH1 for direct fluxes), the solar radiation being then calculated as the sum of the diffuse and direct fluxes. A sky imager (Cloudcam, CMS) is a camera equipped with a fisheye lens to cover the entire upper half sphere. The cloud cover is estimated from an algorithm, named Findclouds and provided by the manufacturer, comparing the different values of the red, green and blue components of each pixel of the images taken (Shukla et al., 2016)."

L158 – section is titled "NPF event frequency and Growth rate" but the growth rate is not included in this section.
The title was modified into : "NPF event frequency"

L169, 172 – I would not compare the undefined frequency with the boreal forest, the environments are pretty different. Have the authors other explanation? Maybe the NPF doesn't growth enough (cut off Dp is 15.3nm)? Similarly for the whole text, the authors compare with

pristine boreal forest in many sections, I would recommend compare with other urban environments when possible.

A large part of the published NPF studies are based on observations performed within the boreal forest, which is why we use those studies for comparison. However, we do understand the importance of comparing our results to similar site types. Throughout the manuscript, we also compare our results to NPF observation in urban environments such as Dos Santos et al. (2015) performed in Paris but also observations performed over Chinese megacities or other megacities in Europe (Beijing, Helsinki, London, Leipzig, Copenhagen, Granada, etc..). In particular the paragraph following this one is dedicated to the comparison of event, non-event and undefined frequencies to those observed in Paris.

The undefined events are defined by Dal Maso et al. as fine particles (Dp < 25nm) that does not grow during their existence. The reviewer is right to highlight the fact that these particles may have stopped growing below the minimum cutoff size of 15.7nm so with our measurements we could have missed the growth of those particles.

L197 – I would recommend "Aerosol number size distribution", and I would also recommend to remove the "dry" term for the whole text (if you follow ACTRIS guidelines, it is assumed to be dry and you don't have another "wet" smps)

We corrected the 3.2 section title as suggested by the referee. We are following the ACTRIS standards by using a Nafion membrane up-stream the SMPS. We removed dry from the manuscript and added one sentence in the description of the SMPS:

"The Scanning Mobility Particle Sizer (SMPS) measures particle number size distribution between 15.7-800 nm downstream a Nafion membrane as recommended by ACTRIS standards to keep relative humidity below 40%. "

Figure 3 – These figures don't have the time resolution of the instrument. Please indicate in the text the average you have use for the data or provide the plots with the instrument time resolution.

Reviewer 1 also commented on those figures and we have modified the figure caption and the text accordingly.

" Figure 3: Hourly median particle number size distribution (15.7 nm<Dp<800 nm) observed during NPF event (a), undefined (b) and non-event (c) days in spring and summer from 2017-2020."

Main text: "Median daily contour plots of the particle number size distributions (PNSD) obtained from the SMPS are shown in Figure 3 separately for NPF event (n ≈ 800 PNSD), undefined (n ≈ 2300 PNSD) and non-event (n ≈ 1700 PNSD) days observed during the warm period (only spring and summer). The PNSD were first selected then averaged to one-hour time resolution using median filtering."

L217 – I would suggest "NPF starting time…" and "growth" instead of "Growth"
This was corrected.

L220 – fewer events "starting" in the early morning?
Right.

L221 – Authors sometimes use GR and others GR15-30, please use only one (figures included). As I mentioned before, I would recommend to use always GR and state in the methods section that it always refers to 15-30 nm size range.

Reviewer 1 also noted this annoying switch from GR to GR15.7 -30. The authors carefully went through the whole manuscript to remove the occurrence of GR when it was referring to our results.

L231 – "presence of availability"?
This is a typo. We meant presence or availability. Thanks for pointing it out.

Figure 4 – the number of cases for the GR is the same than for the starting time figure?
Yes, the number of events per month is the same for all this study. Therefore, we did not reproduce the N on Figure 4b.

Lines missing in page 13, but "nm.h-1" again change "." by "·". Temperature "dependence".
All these mistakes were corrected.

Finally, in this same page, the authors points to the importance of biogenic emissions, however, the measurements were done in an urban environment probably where higher influence of anthropogenic organic compounds are expected? As I pointed in my major comment, what is the contribution of H2SO4? Should be minimal I guess (even more at this size range).
The site is located on the university campus with abundant tree plantings, and about 4 km from the city center. By comparison of several parameters (SSA, scattering coefficient, number concentration etc.) observed over ATOLL to the recent reviews of Laj et al.  and Rose et al. (2021), the site could indeed be considered as close to urban sites. However, the various tree varieties can emit large concentration of organics, especially in summertime. The results shown in Figure 2 of this document clearly highlight that organics have a clear role in the growth of freshly nucleated particles, and that sulfate-related particles are less important in case of NPF event days.

Figure 6 – I would recommend showing same periods for figures a) and b), if not the reader cannot use both figures information.  In this sense, include two boxplot figures (one for each period) or combine solar radiation figures on only one. In addition, here the authors use UTC, and before have been using Local Time, please use always the same and indicate it (e.g. L238). Finally, figure caption is not really clear, I would add c) instead of "b) top and bottom".
All the times used in the manuscript and figures are in UTC, and this mistake was corrected in the revised version. We plotted the cloud fraction over the whole day (Figure 3).

[Figure]

*Figure 3 : Cloud fraction observed during event, undefined and non-event days. The red line represents the median while the lower and upper edges of each box represent the 25th and 75th percentiles, respectively. The lower and upper edges of the whisker represent the 10th and 90th percentiles, respectively.*

Results are similar but the authors truly believed that the most relevant plot is the one already in the manuscript. Indeed, the cloud fraction can change quite quickly during the day. In the morning or in the late afternoon it is not unusual to observe stratiform clouds in Lille. For our study, we know that the cloud cover will influence the NPF occurrence but mostly when the NPF events occur (from 09:00 - 14:00). This is why the authors would like to keep the figure as it is.

L255 – space before reference
Done.

L256, 261 – I don't see the link of these reasons with the data shown on Fig 7. RH<40%, these relative humidity values are not observed at your site. High RH limit some VOC ozonolysis, but what about H2SO4–water nucleation? The authors linked in previous section the GR with the increase of temperature, what about the other parameters?

Within the instrument the RH is always lower than 40% according to ACTRIS standards. However, the RH is also monitored via a meteorological station shown in Figure 7. One can clearly see on this figure, that the event days are associated with lower RH for both seasons in comparison to non-event days. This effect has been observed in many other studies (Bousiotis et al., 2021; Dai et al., 2022; Lv et al., 2018; Baalbaki et al., 2021) performed within real atmosphere. The reasons might be those cited within the manuscript.

Some laboratory measurements performed over a large range of RH values (Yu et al., 2017) observed that the growth rate of sub 3nm particles might be more important (1.5 nm h$^{-1}$) by a factor of almost 2 at high RH (80%) in comparison to low RH (10%). However, according to our result the growth of UFP (below 30nm) might be due to organic vapors.

Based on the results obtained in the CLOUD chamber, RH is expected to enhance the particle formation rate of the H2SO4-water binary mechanism (Duplissy et al., 2016). At ATOLL, we do not measure $H_2SO_4$ so we do not know how much $H_2SO_4$–water nucleation is involved in the first phase of the NPF events observed. However, according to the work of Dunne et al. (2016), we do not expect that this binary nucleation mechanism would play a central role at our site, since it is a priori not very efficient in the boundary layer.

Other parameters (NO2, ozone, Absorption and Scattering Angstrom Exponent, aerosol chemical composition, effective diameter, etc… ) were investigated to better understand the GR variation as a function of temperature. Some expected tendencies were observed such as more BCwb when the temperature is low. However, we did not find any other parameter to better explain the GR variation with temperature. This might be related to the emission of Organic precursors that is not yet measured over ATOLL.

To answer to both comments ($H_2SO_4$–water nucleation and GR increase with temperature), precursor measurements are needed. The deployment of instruments allowing the identification of the nature of the species involved in the process is planned on the site, and will make it possible to answer these questions in the future.

Figure 7 – please use same scale for both periods. Again UTC.
As previously said all the time are in UTC. The scales have been modified.

L273, 279 – It is not clear what are you comparing. For event days are larger, maybe yes, but I can not see the overall differences, just the diurnal evolution. Provide the mean or median values, the period of time that you are averaging, …
In this paragraph, the authors are just describing the CS values observed throughout the day. Indeed, the hourly averaged CS values are always larger than 0.02 s$^{-1}$. We compared this value with previous studies over urban and pristine areas.
We modified this sentence into : "Hourly averaged CS values are high (larger than 2 x 10$^{-2}$ s$^{-1}$) during event days occurring during spring and summer (Figure 8a)."

L334 – Where can we see that?
We did not present any analysis on the undefined events in this manuscript because we were focusing on the conditions favorable for NPF events.

L335, 338 – all the undefined cases show a growth that is stopped? What happen the rest of the percentage?
Yes all undefined events show a growth that stopped. The undefined cases were defined as Dal Maso et al. decision tree suggested. For instance, some particles smaller than 25nm can be observed for more than one hour, but those particles are not growing so the particles remain below 25nm. As the sizes below 15.7nm were not scanned, the particles may have grown from

smaller sizes to 15.7 and then the growth stopped but we do not know about that. These events would be classified as non-event days. For diameters larger than 15.7nm, the undefined days usually exhibit a burst of small particle concentration that disappears during the afternoon as shown in Figure 3b.

A more thorough analysis of these undefined events would be required to investigate the other reasons behind the occurrence of those undefined events. We have not performed that analysis yet and it is beyond the scope of this manuscript.

L341, 342 – almost have not talk about HYSPLIT before, I would introduce here why you use it and the objective. Time, UTC or Local?

As mentioned before, the time is always in UTC. The HYSPLIT local time has been transformed into UTC time. The interest of HYSPLIT was evidenced through the undefined events since we observed that the undefined events are quite often associated with a wind direction shift. Then we introduce HYSPLIT by this sentence: "The shift of the wind orientation leading to a stop of the particle growth indicates that NPF events are associated with certain wind directions or air mass origins. To investigate this, HYSPLIT back trajectories were first sorted as a function of event, non-event and undefined days."

L347, 351 – I am not totally agree with this. CS seems that doesn't play an important role to inhibit the formation of new particles and probably is more the absence of precursor vapors and/or photochemistry (polluted Beijing: Kulmala et al. 2017, Du et al. 2022 or even strong dust events: Nie et al. 2014, Casquero-Vera et al 2020). Are you comparing only clear sky days (no clouds)?

Over ATOLL, the number of dust events was really low but still observable (2-3 dust event per year) and when it occurred, they were clearly characterized by larger $PM_{2.5}$ and $PM_{10}$ concentrations and clearly associated with non-event days.
In these sentences, we wanted to highlight that according to the backtrajectory paths, the air mass could be either enriched or depleted of primary precursors.
The sentence was corrected into: 'Those air masses might then have been slightly enriched in primary precursor vapors. This result is consistent with previous results showing "cleaner" air masses are associated with NPF event cases observed during spring.'

For this analysis, we didn't check the cloud coverage over the whole domain but we observed it at ATOLL. The non-event days, as shown in figure 6a, mostly occur while the cloud coverage is between 0.6 and 0.9. There are a very limited number of non-event (13%) occurring during clear sky (cloud fraction lower than 0.4) conditions throughout the period between 09:00 – 14:00. The study would not have been statistically correct by excluding more than 70% of the data set. We are therefore not comparing only clear sky days.

L373 - NSF6-100 use subindex.
Done.

Section 3.7 – have the authors look the nucleation strength factor for the 50-100 nm size range? It could maybe be an estimation of the increase of CCN due to NPF…??
As suggested, we plotted the diel profiles of $NSF_{50-100}$ for both seasons (MAM and JJA). The $NSF_{50-100}$ values are ranging from 1 to 1.6. The maximum values are reached during the

afternoon at 16:00 and 15:00 during spring and summer, respectively. Of course, the values are not as important as $NSF_{15.7-100}$ but it still highlights the large impact of the NPF events over ATOLL. Indeed, the concentration of CCN-like particles ($50 < Dp < 100nm$) shows an increase during the early afternoon up to a factor of 0.7 (0.3) during summer (spring). This impact may have a large influence on the CCN concentration available for activation. This point needs to be further studied with a CCN counter to evaluate the hygroscopicity of those particles.

Part of this discussion was added within the manuscript:

"As previously shown(Sebastian et al., 2021), NPF events can also play a major role on Earth's radiative budget when the newly formed particles grow to climate-relevant sizes (50-100nm). In order to understand the NPF influence on these particles the $NSF_{50-100}$ was also calculated (see supplementary figures). The results show a large increase up to 1.6 of the $NSF_{50-100}$ in the early afternoon for both seasons. This suggest a potential impact on the CCN concentration that needs to be further studied."

[Figure]

*Figure 4: (a) Diel variation of the Nucleation Strength Factor (NSF50-100) during MAM and JJA calculated from number concentration during the 2017-2020 period.*

References

Baalbaki, R., Pikridas, M., Jokinen, T., Laurila, T., Dada, L., Bezantakos, S., Ahonen, L., Neitola, K., Maisser, A., Bimenyimana, E., Christodoulou, A., Unga, F., Savvides, C., Lehtipalo, K., Kangasluoma, J., Biskos, G., Petäjä, T., Kerminen, V.-M., Sciare, J., Kulmala, M., 2021. Towards understanding the characteristics of new particle formation in the Eastern Mediterranean. Atmospheric Chemistry and Physics 21, 9223–9251. https://doi.org/10.5194/acp-21-9223-2021

Casquero-Vera, J. A., Lyamani, H., Dada, L., Hakala, S., Paasonen, P., Román, R., Fraile, R., Petäjä, T., Olmo-Reyes, F. J. and Alados-Arboledas, L.: New particle formation at urban and high-altitude remote sites in the south-eastern Iberian Peninsula, Atmos. Chem. Phys., 20(22), 14253–14271, doi:10.5194/acp-20-14253-2020, 2020.

Dada, L., Chellapermal, R., Buenrostro Mazon, S., Paasonen, P., Lampilahti, J., Manninen, H.E., Junninen, H., Petäjä, T., Kerminen, V.-M., Kulmala, M., 2018. Refined classification and characterization of atmospheric new particle formation events using air ions (preprint). Aerosols/Field Measurements/Troposphere/Physics (physical properties and processes). https://doi.org/10.5194/acp-2018-631

Dada, L., Paasonen, P., Nieminen, T., Buenrostro Mazon, S., Kontkanen, J., Peräkylä, O., Lehtipalo, K., Hussein, T., Petäjä, T., Kerminen, V. M., Bäck, J. and Kulmala, M.: Long-term analysis of clear-sky new particle formation events and nonevents in Hyytiälä, Atmos. Chem. Phys., 17(10), 6227–6241, doi:10.5194/acp-17-6227-2017, 2017.

Dai, L., Zhao, Y., Zhang, L., Chen, D., Wu, R., 2022. Particle number size distributions and formation and growth rates of different new particle formation types of a megacity in China. Journal of Environmental Sciences. https://doi.org/10.1016/j.jes.2022.07.029

Dameto de España et al. 2017: Long-term quantitative field study of New Particle Formation (NPF) events as a source of Cloud Condensation Nuclei (CCN) in the urban background of Vienna, Atmos. Environ., 164, 289-298.

Du, W., Cai, J., Zheng, F., Yan, C., Zhou, Y., Guo, Y., Chu, B., Yao, L., Heikkinen, L. M., Fan, X., Wang, Y., Cai, R., Hakala, S., Chan, T., Kontkanen, J., Tuovinen, S., Petäjä, T., Kangasluoma, J., Bianchi, F., Paasonen, P., Sun, Y., Kerminen, V.-M., Liu, Y., Daellenbach, K. R., Dada, L. and Kulmala, M.: Influence of Aerosol Chemical Composition on Condensation Sink Efficiency and New Particle Formation in Beijing, Environ. Sci. Technol. Lett., 9(5), 375–382, doi:10.1021/acs.estlett.2c00159, 2022.

Duplissy, J., Merikanto, J., Franchin, A., Tsagkogeorgas, G., Kangasluoma, J., Wimmer, D., Vuollekoski, H., Schobesberger, S., Lehtipalo, K., Flagan, R.C., Brus, D., Donahue, N.M., Vehkamäki, H., Almeida, J., Amorim, A., Barmet, P., Bianchi, F., Breitenlechner, M., Dunne, E.M., Guida, R., Henschel, H., Junninen, H., Kirkby, J., Kürten, A., Kupc, A., Määttänen, A., Makhmutov, V., Mathot, S., Nieminen, T., Onnela, A., Praplan, A.P., Riccobono, F., Rondo, L., Steiner, G., Tome, A., Walther, H., Baltensperger, U., Carslaw, K.S., Dommen, J., Hansel, A., Petäjä, T., Sipilä, M., Stratmann, F., Vrtala, A., Wagner, P.E., Worsnop, D.R., Curtius, J., Kulmala, M., 2016. Effect of ions on sulfuric acid-water binary particle formation: 2. Experimental data and comparison with QC-normalized classical nucleation theory: BINARY PARTICLE FORMATION EXPERIMENTS. J. Geophys. Res. Atmos. 121, 1752–1775. https://doi.org/10.1002/2015JD023539

Kulmala, M., Kerminen, V. M., Petäjä, T., Ding, A. J. and Wang, L.: Atmospheric gas-to-particle conversion: Why NPF events are observed in megacities?, Faraday Discuss., 200, 271–288, doi:10.1039/c6fd00257a, 2017.

Lehtipalo, K., Yan, C., Dada, L., Bianchi, F., Xiao, M., Wagner, R., Stolzenburg, D., Ahonen, L.R., Amorim, A., Baccarini, A., Bauer, P.S., Baumgartner, B., Bergen, A., Bernhammer, A.-K., Breitenlechner, M., Brilke, S., Buchholz, A., Mazon, S.B., Chen, D., Chen, X., Dias, A., Dommen, J., Draper, D.C., Duplissy, J., Ehn, M., Finkenzeller, H., Fischer, L., Frege, C., Fuchs, C., Garmash, O., Gordon, H., Hakala, J., He, X., Heikkinen, L., Heinritzi, M., Helm, J.C.,

Hofbauer, V., Hoyle, C.R., Jokinen, T., Kangasluoma, J., Kerminen, V.-M., Kim, C., Kirkby, J., Kontkanen, J., Kürten, A., Lawler, M.J., Mai, H., Mathot, S., Mauldin, R.L., Molteni, U., Nichman, L., Nie, W., Nieminen, T., Ojdanic, A., Onnela, A., Passananti, M., Petäjä, T., Piel, F., Pospisilova, V., Quéléver, L.L.J., Rissanen, M.P., Rose, C., Sarnela, N., Schallhart, S., Schuchmann, S., Sengupta, K., Simon, M., Sipilä, M., Tauber, C., Tomé, A., Tröstl, J., Väisänen, O., Vogel, A.L., Volkamer, R., Wagner, A.C., Wang, M., Weitz, L., Wimmer, D., Ye, P., Ylisirniö, A., Zha, Q., Carslaw, K.S., Curtius, J., Donahue, N.M., Flagan, R.C., Hansel, A., Riipinen, I., Virtanen, A., Winkler, P.M., Baltensperger, U., Kulmala, M., Worsnop, D.R., 2018. Multicomponent new particle formation from sulfuric acid, ammonia, and biogenic vapors. Science Advances 4, eaau5363. https://doi.org/10.1126/sciadv.aau5363

Lv, G., Sui, X., Chen, J., Jayaratne, R., Mellouki, A., 2018. Investigation of new particle formation at the summit of Mt. Tai, China. Atmos. Chem. Phys. 18, 2243–2258. https://doi.org/10.5194/acp-18-2243-2018

Marten, R., Xiao, M., Rörup, B., Wang, M., Kong, W., He, X.-C., Stolzenburg, D., Pfeifer, J., Marie, G., Wang, D. S., Scholz, W., Baccarini, A., Lee, C. P., Amorim, A., Baalbaki, R., Bell, D. M., Bertozzi, B., Caudillo, L., Chu, B., Dada, L., Duplissy, J., Finkenzeller, H., Carracedo, L. G., Granzin, M., Hansel, A., Heinritzi, M., Hofbauer, V., Kemppainen, D., Kürten, A., Lampimäki, M., Lehtipalo, K., Makhmutov, V., Manninen, H. E., Mentler, B., Petäjä, T., Philippov, M., Shen, J., Simon, M., Stozhkov, Y., Tomé, A., Wagner, A. C., Wang, Y., Weber, S. K., Wu, Y., Zauner-Wieczorek, M., Curtius, J., Kulmala, M., Möhler, O., Volkamer, R., Winkler, P. M., Worsnop, D. R., Dommen, J., Flagan, R. C., Kirkby, J., Donahue, N. M., Lamkaddam, H., Baltensperger, U. and El Haddad, I.: Survival of newly formed particles in haze conditions, Environ. Sci. Atmos., 2(3), 491–499, doi:10.1039/D2EA00007E, 2022.

Nie, W., Ding, A., Wang, T., Kerminen, V. M., George, C., Xue, L., Wang, W., Zhang, Q., Petäjä, T., Qi, X., Gao, X., Wang, X., Yang, X., Fu, C. and Kulmala, M.: Polluted dust promotes new particle formation and growth, Sci. Rep., 4(1), 6634, doi:10.1038/srep06634, 2014.

Rose, C., Collaud Coen, M., Andrews, E., Lin, Y., Bossert, I., Lund Myhre, C., Tuch, T., Wiedensohler, A., Fiebig, M., Aalto, P., Alastuey, A., Alonso-Blanco, E., Andrade, M., Artíñano, B., Arsov, T., Baltensperger, U., Bastian, S., Bath, O., Beukes, J. P., Brem, B. T., Bukowiecki, N., Casquero-Vera, J. A., Conil, S., Eleftheriadis, K., Favez, O., Flentje, H., Gini, M. I., Gómez-Moreno, F. J., Gysel-Beer, M., Hallar, A. G., Kalapov, I., Kalivitis, N., Kasper-Giebl, A., Keywood, M., Kim, J. E., Kim, S.-W., Kristensson, A., Kulmala, M., Lihavainen, H., Lin, N.-H., Lyamani, H., Marinoni, A., Martins Dos Santos, S., Mayol-Bracero, O. L., Meinhardt, F., Merkel, M., Metzger, J.-M., Mihalopoulos, N., Ondracek, J., Pandolfi, M., Pérez, N., Petäjä, T., Petit, J.-E., Picard, D., Pichon, J.-M., Pont, V., Putaud, J.-P., Reisen, F., Sellegri, K., Sharma, S., Schauer, G., Sheridan, P., Sherman, J. P., Schwerin, A., Sohmer, R., Sorribas, M., Sun, J., Tulet, P., Vakkari, V., van Zyl, P. G., Velarde, F., Villani, P., Vratolis, S., Wagner, Z., Wang, S.-H., Weinhold, K., Weller, R., Yela, M., Zdimal, V., and Laj, P.: Seasonality of the particle number concentration and size distribution: a global analysis retrieved from the network of Global Atmosphere Watch (GAW) near-surface observatories, Atmos. Chem. Phys., 21, 17185–17223, https://doi.org/10.5194/acp-21-17185-2021, 2021

Yu, H., Dai, L., Zhao, Y., Kanawade, V.P., Tripathi, S.N., Ge, X., Chen, M., Lee, S.-H., 2017. Laboratory observations of temperature and humidity dependencies of nucleation and growth rates of sub-3 nm particles. Journal of Geophysical Research: Atmospheres 122, 1919–1929. https://doi.org/10.1002/2016JD025619

Yus-Díez, J., Via, M., Alastuey, A., Karanasiou, A., Minguillón, M. C., Perez, N., Querol, X., Reche, C., IvanÄ iÄ , M., Rigler, M., and Pandolfi, M.: Absorption enhancement of black carbon particles in a Mediterranean city and countryside: effect of particulate matter chemistry, ageing and trend analysis, Atmos. Chem. Phys., 22, 8439–8456, https://doi.org/10.5194/acp-22-8439-2022, 2022.

---

## Author Response (AR2)

We would like to thank the Editor for his comments and his suggestions that have improved the quality of the manuscript. The first author would like to apologize. The version that was uploaded on the 27 of october was not the final one which explain the numerous mistakes by the editor. Over the past few days, several authors have been working to improve the English as well as the figures. All the editor corrections were taken into account as suggested in the main text but also in the supplementary materials.